# A pre-calibration approach to select optimum inputs for hydrological models in data-scarce regions

Esraa Tarawneh[1], Jonathan Bridge[1] and Neil Macdonald[2]

[1]Centre for Engineering Sustainability, School of Engineering, University of Liverpool, Liverpool, L69 3GH, UK

[2] School of Environmental Sciences, University of Liverpool, Liverpool, L69 7ZT, UK

*Correspondence to*: Esraa Tarawneh (e.tarawneh@liverpool.ac.uk)

**Abstract.** This study uses the Soil and Water Assessment Tool (SWAT) model to quantitatively compare available input datasets in a data-poor dryland environment (Wala catchment, Jordan; 1 743 km$^2$). Eighteen scenarios combining best available land-use, soil and weather datasets (1979 - 2002) are considered to construct SWAT models. Data include local

observations and global reanalysis data products. Uncalibrated model outputs assess the variability in model performance derived from input data sources only. Model performance against discharge and sediment load data are compared using $r^2$, NSE, RSR and PBIAS. NSE statistic varies from 0.56 to -12 and 0.79 to -85 for best and poorest-performing scenarios against observed discharge and sediment data respectively. Global weather inputs yield considerable improvements on discontinuous local datasets, whilst local soil inputs perform considerably better than global-scale mapping. The

methodology provides a rapid, transparent and transferable approach to aid selection of the most robust suite of input data.

**Keywords:** soil; land-use; weather; sediment; sensitivity; SWAT

## 1 INTRODUCTION

Arid and semi-arid regions of the world suffer from water scarcity exacerbated by growing populations, increasing per capita water consumption and agricultural intensification. Depletion of surface water and over-abstraction of non-renewable

groundwater adversely impact ecosystems and human quality of life (Wheater et al., 2008b). Effective water management is crucial and relevant decision making can be assisted by approximating the complex hydrologic systems of arid and semi-arid regions through modelling. This enables scenario-testing and forecasting to inform decision-making in water and land management (Tessema, 2011; Wheater et al., 2008a).

The ability of a model to successfully predict catchment behaviour relies on the reliability and representativeness of

the data against which it is calibrated, the quality of the processes and parameters assumed internally within the model and the accuracy of the input datasets used to define the catchment (Griensven and Meixner, 2006). Unfortunately in many cases, the regions most in need of reliable hydrological models are those with limited economic resources and fragmented environmental monitoring infrastructure (Ragab and Prudhomme, 2002). Data available to underpin models may, therefore, vary significantly, both in quality and quantity (Pilgrim et al., 1988). This encourages the use of modelling 'rules of thumb'

or estimations based on spatially or temporally aggregated data for the modelled area, or data obtained from comparable better-studied regions (Gee and Hillel, 1988; Nyong et al., 2007; Tingsanchali and Gautam, 2000).

Imperfect fit of model results to measured data is called modelling uncertainty while predictive uncertainty measures predictability of the model when used for future scenarios (Krupnick et al., 2006). Structure of hydrologic models
can lead to uncertainty issues, particularly when assumptions are inherent within the model design. However, choices available to semi-arid regions are still limited and reducing associated uncertainty requires intensive research to improve incorporated mathematical models and their ability of representing physical processes and extract information from available data. Uncertainty related to measurements used for model assessment can only be reduced by improving observation techniques and networks (Griensven and Meixner, 2006). This study focuses on modelling uncertainty and particularly on
inputs as one of its main sources in models of data-sparse semi-arid regions. Minimizing inputs uncertainty is an important aspect of the planning process for modelling projects, it ensures that input data and parameters are more accurate and suitable, reduces predictive uncertainty and assists decision-making (Liu and Gupta, 2007; USEPA, 2002). Furthermore, in the absence of high quality 'ground truth' data for soils, land use and weather inputs, powerful automated calibration algorithms can alter model parameters to produce a structurally biased model (Kalantari et al., 2015) which provides a good
fit to specified calibration data, but may diverge significantly from true catchment behaviour under other conditions (Beven, 2011).

The relationship between model inputs and performance are investigated at a range of scales in different hydrologic settings (Beeson et al., 2014; Chaplot, 2013; Lobligeois et al., 2013; Lobligeois et al., 2014; Müller Schmied et al., 2014). For example, Legesse et al. (2003) use distributed Precipitation-Runoff Modelling (PRM) to investigate the impact of
climatic and land-use variations on hydrologic response in data-scarce Tropical Africa. Di Luzio et al. (2005) determine that Digital Elevation Model (DEM) construction is critical to stream flow and sediment predictions of a SWAT (Arnold et al., 1998) model for a 21.3 km$^2$ watershed in the Mississippi, with a significant effect of land-use and limited influence of soil data. Liong et al. (2013) present SWAT model results for a catchment in Southeast Asia and conclude that the highest uncertainty results from applying global climate models for regional and localized applications, recommending the use of
higher spatial resolution regional data. Recently, Faramarzi et al. (2015) show in a SWAT analysis of Alberta, Canada that choice of optimal input datasets significantly affects the overall model performance by reducing unnecessary and arbitrary adjustment of parameters to compensate for structural errors in the model. Crucially, better model performance is not necessarily correlated with accumulation of a mass of data; rather it depends on data reliability and relevance (Tessema, 2011).

In settings where input and output datasets are robust and comprehensive, such as in humid areas, this issue may present rarely or be mitigated by transfer of knowledge from neighbouring or geomorphically similar catchments. By contrast, in semi-arid regions, for example, where data coverage and quality are historically poor (Edmunds et al., 2013) and hydrological systems operate under significantly different conditions from those in well-monitored temperate environments (Chehbouni et al., 2008), transfer of parameters may not be the best option and can be  itself a source of uncertainty

(Wheater et al., 2008a). Therefore, developing methodologies that target optimizing data and parameters of the area itself, in addition to improving observation techniques and networks are more efficient ways to reduce uncertainty (Griensven and Meixner, 2006). When the relative integrity of available datasets is unknown and research resources are limited, the questions arise: which dataset(s) should be employed in modelling, and where should investment be targeted to improve data quality?

This study explores a methodology for differentiating between various input datasets of unknown relative quality for a hydrological model of a typical semi-arid catchment in Jordan. We start with the proposition that the combination of input datasets which produce the best fit to observed output data prior to full model calibration will yield a model that is less computationally intensive and which minimises the potential for structural errors arising from systematic biases introduced during calibration. Our objective is to test the specific hypothesis that different combinations of a small number of available datasets will result in a significant variation in pre-calibration model performance, allowing rapid estimation of relative input data quality. The aim is to develop a simple, resource-efficient and transferable method for use in the design and specification of catchment models to support water resource management where data is of uncertain quality and/or quantity and decisions on where to invest efforts to improve them are limited by available resources.

## 2 STUDY AREA

Jordan is one of the poorest countries globally in terms of water resources and availability with less than 200 mm annual rainfall across 91% of its area (Abdulla and Al-Assa'd, 2006). Hence, severe water stress and the ongoing unsustainable drawdown of fossil groundwater reserves in Jordan make pilot schemes for increasing the capture of seasonal storm flows of considerable strategic importance. The Wala basin forms the northern 2 100 km$^2$ of the Mujib basin in central Jordan (Fig. 1). Its main drainage stream is Wadi Wala, which flows from an elevation 750 m to -100 m (a.s.l), where it joins Wadi Mujib and their confluence flows to the Dead Sea (Cordova, 2008). It has a Mediterranean climate characterized by hot dry summers and cold wet winters (Al-Bakri and Al-Jahmany, 2013). Maximum precipitation occurs in December and January while the rainy season extends between October and May. Average annual rainfall decreases in a northwest-southeast gradient from 500 mm a$^{-1}$ to less than 100 mm a$^{-1}$, with an average of 181 mm a$^{-1}$ (Margane et al., 2009).

The area of this study, the Wala catchment, occupies 1 743 km$^2$ upstream of the Wala Dam (Fig. 1). The Wala catchment and the aquifer beneath it form an important hydrologic system in Jordan, with the Wala Dam (31.56 ºN, 35.80 ºE) constructed between 1999 and 2002 to artificially recharge groundwater storage. This recharge supports agricultural activities in downstream cultivated areas as well as supplementing the potable water needs of the capital city Amman via abstraction wells approximately 9 km downstream of the dam at Al-Heidan (Ta'any, 2011). Wadi Wala had a permanent discharge before intense pumping started in the 1990s (Cordova, 2008). The main agricultural activity within the catchment is sheep and goat grazing, and its land cover is characterized by scrub vegetation, minor tree cover and some irrigated and non-irrigated crops. Since plans are in place for an expansion in the number of artificial recharge schemes, funded by UN

and other international aid monies (JNFP, 2012; Margane et al., 2009), Wala provides a critical and influential case study in the development and management of catchment water resources in Jordan and the wider region.

## 3 METHODS

### 3.1 Approach

All available weather, soil, land-use and topographic datasets for the catchment are collated and characterised as described below. The general form and boundary conditions of the catchment are implemented in the SWAT (ArcSWAT 2012, http://swat.tamu.edu/software/arcswat/) model framework and used as input factorial combinations of the available datasets, yielding 18 different model representations of the Wala catchment. These models are run prior to internal calibration in order to elucidate the range of input influences on model performance, using observed discharge and empirical sediment data as benchmarks for comparison of model outputs. Visual and statistical assessment criteria are applied to check goodness of fit of scenarios outputs and observations both visually and quantitatively.

### 3.2 Model selection and structure

SWAT is selected for this work as it enables a continuous real-time model to simulate hydrology, land management and sedimentation processes on a basin-wide scale (Arnold et al., 1998; Srinivasan et al., 1998). SWAT also uses physical data for topography, weather, soil properties and land-use to directly simulate physical processes, rather than depending on regression formulas to determine input-output relationships (Arnold et al., 2013). Model parameters, input variables and methods that pertain to each type of the main inputs discussed in this study are detailed by Neitsch et al. (2011). There is extensive literature on SWAT and its applications in general (Arnold and Fohrer, 2005; Arnold et al., 2012) and in dryland in particular (see for example Adem et al. (2016); Adham et al. (2016); Havrylenko et al. (2016); Özcan et al. (2016)). The applicability of SWAT in arid environments is assessed by Wu et al. (2016) and the results obtained encourage using SWAT in regions of similar characteristics. Zhang et al. (2016) show by comparing the performance of a conceptual model and SWAT, as a physically-based model, in simulating arid regions in China that both models perform reasonably well; however, data constraints and deficiencies, if not addressed properly, can limit SWAT's performance. In another comparison of modelling tools capability of simulating arid regions undertaken by Liu et al. (2016), SWAT shows strengths in different hydrological processes important to study arid regions such as lateral flow, while it performs relatively poorer in other processes, which are not main interest in this study such as snow, hence the outcome supports our selection. Marek et al. (2016) investigate evapotranspiration in the semi-arid Texas High Plains and state that SWAT is a suitable tool to simulate it. Shrestha et al. (2016) demonstrate the capability of SWAT to perform realistically in areas of extreme conditions like the semi-arid Onkaparinga catchment, South Australia and stress on the value of improving data sources for more realistic performance and robust simulation using SWAT. Several hydrological modelling studies undertaken in arid and semi-arid regions in Tunisia indicate that SWAT simulates various hydrological processes of these areas with reasonable accuracy and

reliability (Ouessar et al., 2009; Ouessar et al., 2008). In Jordan, and adjacent to the Wala catchment, SWAT is employed successfully by Ijam and Al-Mahamid (2012) to estimate sediment inflow to the Mujib Dam reservoir and identify patterns of soil erosion across the Mujib basin. However, their study strongly recommends improving field measurements of sedimentation in the area for more confidence in the proposed model in simulating sediments as the data utilized for model calibration are constructed based on previous studies to cover the shortage in observed data. Ijam and Tarawneh (2012) also predict water and sediment yields from the Wala catchment using SWAT and conclude that it satisfactorily simulates hydrological processes and sedimentation in the area but again stress on the obstacles posed by the lack of data, especially those required for model calibration, the case in which reducing input uncertainty may partially account for potential errors. The key features of the SWAT model are briefly described below.

SWAT applies two levels of physical discretization: i) watershed into subbasins; and, ii) subbasins into hydrologic response units (HRUs), which are regions of unique soil, slope and land-use combinations (Arnold et al., 1998; Srinivasan et al., 1998). SWAT employs input weather information along with water budget techniques to quantitatively describe interrelated watershed hydrology components on a daily basis (Betrie et al., 2011). The model applies the US Soil Conservation Service Curve Number (CN) method to transform daily rainfall to surface runoff (USDA, 1972) while the CN varies according to basin characteristics including the type of land-use, soil group and antecedent moisture content (USDA, 1986). These parameters are extracted/calculated from the soil and land-use data. A set of widely tested sub-models are incorporated into SWAT to simulate key hydrological functions:

- the Rational method (Chow et al., 1988) to predict peak discharge rate depending on daily precipitation, calculated surface runoff and topographic parameters derived from the DEM;
- crack-discharge model combined with storage routing techniques and direct soil parameters such as hydraulic conductivity, percent clay content, available water capacity and bulk density to estimate percolation (Arnold et al., 1995);
- the Penman-Monteith method (Monteith, 1964) for evapotranspiration, which depends on daily wind speed; maximum/minimum temperature, evaporative demand of soil and characteristics of land-cover leaves. These are all provided to the model through weather, soil and land-use data;
- the variable storage coefficient to compute channel discharge routing (Williams, 1969), for which length, slope and Manning's value of channels are important information derived from the DEM and soil data (Neitsch et al., 2011);
- the Williams and Singh (1995) formulas introduced for the Universal Soil Loss Equation (USLE) and its modification (MUSLE) to predict gross soil erosion and sediment yield at HRU-level respectively. Key parameters required for these formulas are soil erodibility factor, percent rock in soil, cover/ management and practice factors taken from land-use data, topographic factor and parameters of surface runoff and peak discharge as detailed clearly in (Neitsch et al., 2011).

The model considers channel degradation as a result of stream energy and sediment deposition in channels, according to particle fall velocity, to investigate sediment transport (Williams, 1980). The input datasets required to represent the physical characteristics of the area and provide the model parameters are described in the following sections. Running the SWAT model yields a range of outputs for different model components, including the watershed, subbasins, HRUs and channel system (see Arnold et al. (2013) for full description).

## 3.3 Catchment configuration

The catchment area is defined by reference to the 30 m-resolution DEM obtained from ASTER-GDEM version 2 (Tachikawa et al., 2011). The DEM is used to derive topographic parameters for the catchment area, such as overland slope and slope length, and define stream pattern according to customized threshold of the area contributing to each branch (Di Luzio et al., 2002). An optimal subbasin area threshold of 3 % (5000 ha for Wala) of watershed area (Jha et al., 2004) resulted in 23 model subbasins, for which main streams and outlets are defined (Fig. 2). The main streams of subbasins 15 and 19 form the arms of the Wala dam reservoir, and their confluence is the main stream of subbasin 16, representing the catchment outlet and dam location (Tarawneh, 2007).

Land slope is derived from the 30-m DEM described above and shows relatively flat topography over the upper catchment compared to steep canyons near the outlet. Relatively steep slopes characterize the western and northern parts of the watershed. Average and maximum slopes within the area are 5.42° and 54.4°, respectively.

## 3.4 Input data

### 3.4.1 Land use

Land-use maps from three different sources are used: i) a 30 m-resolution raster grid reprocessed from the detailed map developed and presented by Tarawneh (2007) based on the 1:250 000-scale map of the National Soil Map and Land-use Project of Jordan (Ministry of Agriculture, 1994); ii) the land use/cover map of Jordan produced by Al-Bakri et al. (2013) and also reprocessed to a 30 m-resolution grid; iii) the Europe/Asia land-use grid (WaterBase, 2012) constructed from the Global Land Cover Characterization (GLCC) database with a 1:2 000 000 scale and 1 km spatial resolution. Figure 3 illustrates that the three maps all show two dominant types of vegetation over the area, with minor coverage by other land-use classes. Considering the importance of land use to modelling and planning of drylands, it is essential to take into consideration any major land use differences over time (Wolffc, 2011). By investigating land use variation in several sites in the Badia zone in Jordan over the period 1953 – 1992, Al-Bakri et al. (2001) conclude that land use changes are from rangeland to cultivated areas and urban settlements in addition to the appearance of some irrigation fields after 1978. For the Wala catchment, land use maps and information from different sources and periods tend to exhibit only minor differences in the major land use types and this supports our assumption that land use changes in the study area can be neglected for the

time frame of the current study (1979 – 2002), particularly that the model considers the major land use type (greater than specified threshold) in each hydrologic response unit.

### 3.4.2 Soil

The physical soil characteristics required by the model include soil hydrologic group (Wood and Blackburn, 1984), depth of soil layers, moist bulk density, available water capacity, saturated hydraulic conductivity, organic carbon content, erodibility factor, moist soil albedo, rock fragment content and percentages of silt, sand and clay (Arnold et al., 2011). Two soil datasets are compared (Fig. 4): i) the Europe/Asia soil grid (WaterBase, 2012) produced from the Food and Agriculture Organization (FAO) map with 1:25 000 000-scale and a coarse spatial resolution of 10 km (Leon, 2011), showing only three types of two-layer soils over the Wala catchment; ii) the map produced by Tarawneh (2007) and processed to 30 m-resolution based on the 1:250 000-scale soil map and analysis released by the Jordanian government (Ministry of Agriculture, 1994). In the latter case, the catchment is divided into 17 three-layer soil units, each linked to a soil properties database based on thorough sampling undertaken by the national project to study soil profile, composition and spatial distribution. The Tarawneh (2007) map provides higher resolution and level of detail and more importantly, measured ground-truth based data including silt, sand and clay percentages, percent organic carbon and rock content, which are used to define soil texture and estimate or calculate the remaining characteristics using pre-developed models, equations or graphs.

### 3.5 Hydrologic response units (HRUs)

Soil, land-use and slope combinations define HRUs, over which water and sediment loadings are estimated. Hence, each set of input data defines a unique set of HRUs and this provides a key structural characteristic that governs sensitivity of the model to changes in these fundamental input datasets. Different combinations of the input datasets specified above result in significant differences in the number and physical characteristics of HRUs and consequently water and sediment loading simulation at both HRU and subbasin levels. For instance, combining the WaterBase (2012) soil data (Fig. 4a) with each of the three land-use maps shown in Fig. 3 results in 47, 67 and 68 HRUs respectively, based on a multiple threshold criteria of 20, 30 and 30 % applied on land use, soil and slope respectively. The number of HRUs generated in each of the different scenarios is displayed alongside modelling results presented later in this paper.

### 3.6 Input weather datasets

SWAT requires daily series of climatic data as model input. Where incomplete climate records exist, SWAT uses a built-in weather generator algorithm to statistically process monthly data taken from representative weather stations to produce full daily series or fill any missing records in the available measured data (Arnold et al., 2013). The SWAT generator uses a first-order Markov chain model to predict wet/dry days depending on monthly wet/dry probabilities provided by the user. Daily precipitation is estimated for wet days using a skewed distribution while a normal distribution is used to generate missing maximum/minimum temperature and solar radiation in conjunction with a continuity equation. These values are adjusted

depending on wet/dry conditions so that the monthly average of generated daily values agrees with the averages provided by the user. Details of the SWAT weather generator are provided by Neitsch et al. (2011). Average monthly climatic parameter data from the Qatraneh (31.24 ºN, 36.04 ºE) and Errabbah (31.27 ºN, 35.74ºE) weather stations (Fig. 5) over ten years are processed to provide two weather generator files. The mentioned stations are located outside the watershed but it is a common practice in watershed modelling to use weather data monitored outside the study area, though some potential complications may arise regarding validity and representativeness of these stations (Fuka et al., 2014).

Daily precipitation records from 26 gauges in and around the study area obtained from the Ministry of Water and Irrigation of Jordan are used. Record lengths vary, with the earliest record starting in 1938. However, detailed analysis of these datasets revealed poor quality and gaps in most of the series, leaving only three gauges of sufficient quality to provide continuous daily records between January 1971 and September 2002. Figure 5 shows the rain gauges used in this study (noting their respective missing record percentage over 31 years). The stations are preferentially distributed to the west of the catchment, as such the representativeness of these stations may be poorer to the east of the catchment. A considerable portion of the Madaba gauge (31.71 ºN, 35.79 ºE) record is missing. The Madaba gauge is used in this study to demonstrate model sensitivity to gaps in rainfall information in the semi-arid region characterised by intense, highly intermittent storms.

Temperature is important for key processes in the hydrologic cycle such as evapotranspiration and vegetation growth (Sandholt et al., 2002). The weather stations used in this study, Qatraneh and Errabbah (Fig. 5) hold records of daily maximum/minimum temperature for the period 1971 - 2002, with infrequent gaps. By reviewing the temperature variation, we found it followed a smoother pattern than that of precipitation, hence easier to estimate or forecast to fill the gaps. Figure 6 shows an illustrative subset of daily precipitation and temperature (maximum and minimum) for the two stations used in the model in the representative period 2000 - 2003. Recent research suggests significant increasing trends in daily maximum and minimum temperatures in the Middle East and north Africa over the last 50 - 100 years (Ageena et al., 2014, 2013) which are directly proportional to increasing aridity (see for example Zhang et al. (2005), Trondalen (2009)). However, since the current model is run for shorter periods, long-term climate change is not considered significant in this study.

Global atmospheric reanalyses such as the Climate Forecast System Reanalysis (CFSR) are routinely used to provide catchment-scale hydrological simulations with the required climatic data, particularly in locations for which measured data are scarce (Wang et al., 2011). The CFSR is designed by the National Centre for Environmental Prediction (NCEP) to provide continuous weather data for grid points across the globe for the period 1979 – 2010 (Saha et al., 2010). In an area such as Wala, characterised by intermittent, intense, often localised rainstorms, it is pertinent to query whether such a global reanalyses can adequately capture the local drivers of hydrological activity. Four of the CFSR data points are in or close to the study area (Fig. 5) and therefore their data (daily precipitation, temperature, solar radiation, relative humidity and wind speed) are used as an additional input dataset to compare with the local weather station data.

## 3.7 Scenario comparison

Three land use maps and two soil maps are combined factorially with three sets of weather data obtained from: i) CFSR; ii) local stations including Madaba; iii) local stations excluding Madaba, yielding 18 different model scenarios (Tables 1 and 2). We compare the average monthly stream outflow (discharge, m3s-1) and the monthly sediment transported out of reaches (t) with observations of discharge and observation-derived sediment yield, respectively. The observed discharge data comprise average monthly discharge (m3s-1) obtained from daily measurements at the Wadi Wala flow station CD0038 (31.55 ºN, 35.77 ºE), located 5 km downstream of the current dam location (Margane et al., 2009) for the period January 1971 to September 2002 (before impoundment started), available from the Ministry of Water and Irrigation of Jordan. Howard Humphreys and Partners (1992) provide a sediment-rating curve identifying a strong log linear relationship between log sediment yield (kg s-1) and log discharge (m3s-1) for the Wala gauging station during the design studies of the Wala Dam. Howard Humphreys and Partners (1992) provide a sediment-rating curve identifying a strong log linear relationship between log sediment yield (kg s$^{-1}$) and log discharge (m$^3$ s$^{-1}$) for the Wala gauging station (CD0038) during the design studies of the Wala Dam. This relationship is presented by Tarawneh (2007) in Eq 1 below and used to develop values of sediment yield (t) corresponding to the available discharge values at station CD0038, which are used in this study. Observed sediment accumulation in Wala Dam since construction (2002-2007) closely relates to that modelling over the coeval period (Tarawneh, 2007; Wala Dam Management, 2013).

$$\log Discharge = 0.5833 \log Sediment\ yield + 0.0168 \tag{1}$$

The SWAT model in this study is setup to produce monthly output by averaging daily estimates to simulate seasonal variation of discharge and sediment in the period January 1979 through to January 2003. Model performance under each scenario (combination of input datasets) is evaluated by quantitative comparison with the observed discharge and observation-derived sediment load data series using standard graphical and statistical techniques for watershed modelling (Moriasi et al., 2007). Hydrograph comparison (Yen, 1995) of simulated and observed discharge and sediment yield are combined with quantitative measures.

A suite of four standard statistical instruments are employed to compare input scenarios on the basis of pre-calibration modelled vs observed catchment outputs: coefficient of determination $r^2$ (Eq.2) (Goodwin and Leech, 2006):

$$r^2 = \left( \frac{\sum_{i=1}^{n}(O_i - \bar{O})(P_i - \bar{P})}{\sqrt{\sum_{i=1}^{n}(O_i - \bar{O})^2 \sum_{i=1}^{n}(P_i - \bar{P})^2}} \right)^2 \tag{2}$$

where O is observed and P is predicted values; Nash-Sutcliffe Efficiency (NSE) (Eq. 3) developed by Nash and Sutcliffe (1970):

$$NSE = 1 - \frac{\sum_{i=1}^{n}(O_i - P_i)^2}{\sum_{i=1}^{n}(O_i - \bar{O})^2} \tag{3}$$

root mean square error standard deviation ratio (RSR) (Eq. 4):

$$RSR = \frac{\sqrt{\sum_{i=1}^{n}(O_i - P_i)^2}}{\sqrt{\sum_{i=1}^{n}(O_i - \overline{O})^2}} \tag{4}$$

and percent bias (PBIAS) (Eq. 5) (Moriasi et al., 2007):

$$PBIAS = \frac{\sum_{i=1}^{n}(O_i - P_i)*100}{\sum_{i=1}^{n}O_i} \tag{5}$$

5 Whilst input data (climatic) are based on a daily temporal scale, the model outputs are considered at a monthly timescale for several reasons, i) daily observations of discharge and sediment are unavailable at the Wala station for the whole period of study, with only monthly observations available for model evaluation; ii) A shorter period (1990 – 1996) of daily observations are available at Wala station, but using these yields poor correlations <0.1 between daily model-simulated and observed discharge, iii) with incomplete/low quality measurements, potential for lag within the pairs of daily simulated and
10 observed values (for model statistical evaluation) can present challenges, which can be reduced when using aggregated temporal data; and most importantly, iv) the objective of this study is to determine long-term flux within the catchment, avoiding the complexity presented by ephemeral systems and since the monthly comparison achieves reasonable fit between observed and simulated values, it is considered sufficient for evaluation of the current model with more convenience. However, all calculations of the model occur on a daily time step, which ensures that hydrological events are accounted for
15 separately as they occur each day.

   A similar approach is adopted in several comparable studies, particularly using SWAT, in both humid and arid regions. Spruill et al. (2000) evaluate daily and monthly SWAT models simulation for a small watershed in central Kentucky and state that SWAT is an efficient tool for monthly runoff simulation with NSE values of 0.58 – 0.89 compared to - 0.04 – 0.19 for daily runoff simulation during the same period. The reason suggested is that the model poorly detects peak flows
20 and recession rates while it performs better with total monthly values. For reasonable performance of SWAT, Huang and Zhang (2004) select to simulate discharge in a semi-arid catchment in China on a monthly basis, which leads to NSE of 0.88. The difference between daily and monthly simulations is investigated in watersheds of different scales by Heathman and Larose (2007) and the results show that simulating higher discharge rates, which is usually associated with larger watersheds, introduces greater uncertainty in SWAT discharge estimates and the study states that very good model
25 performance is achieved for monthly stream-flow estimation while the outputs of daily simulation are only within acceptable range.

**4 RESULTS AND DISCUSSION**

**4.1 Comparison of statistical measures**

All four statistics exhibit significant variability in model behaviour among the eighteen input scenarios. Figures 7 and 8 show the $r^2$ and PBIAS statistics, respectively, for each scenario. For discharge prediction, highest $r^2$ is obtained from
scenarios 16, 10 and 4 (group 1); and a far lower $r^2$ is associated with scenarios 2, 8 and 14 (group 3). Values of $r^2$ for the remaining twelve scenarios (group 2) are located between these two groups, with slight or no difference between successive scenarios. $r^2$ values for sediment prediction show higher correlation than that of predicted discharge (Fig. 7b). It should be noted that $r^2$ quantifies only the dispersion; therefore in some cases very good $r^2$ values may be obtained when the model is over/under-predicting all the time regardless of the accuracy. For PBIAS, Fig. 8a shows that scenarios 16, 10 and 4 (which
use the CFSR data) tend to underestimate discharge, while all remaining scenarios show overestimation. Figure 8b shows that scenarios 16, 10 and 4 have least tendency to over/under-predict sediment yield with PBIAS values of 31, -17 and -33, respectively, while all other scenarios significantly overestimate sediment yield. Both indicators consistently identify the input scenarios that most closely represent the observed discharge and sediment data prior to calibration, but yield little further information with which to differentiate between scenarios.

15          NSE and RSR enable a finer distinction between scenarios, revealing clear trends arising from the influence of the different input datasets (Table 1 and Table 2). Table 1 shows the 18 scenarios arranged according to NSE and RSR, with similar descending order for both criteria and a clear structure evident in the importance of different inputs. Scenarios are divided into two distinct groups: i) those using the CFSR dataset, and ii) those applying a combination of generated and locally measured series of weather variables. NSE drops and RSR increases significantly for equivalent scenarios when only
the weather varies (e.g. scenario pairs 16/18 and 1/2), with improved performance for the CFSR in all cases. Several studies worldwide lead to similar results and show that CFSR data out-perform local records. Potential causes for this focus on the data incompleteness, data quality, representativeness of instrumental site data to wider regions, data management and instrument maintenance being common factors; these present a number of challenges, particularly in areas and regions with challenging climates (Wheater et al., 2008b). With special reference to SWAT, Fuka et al. (2014) states that providing
SWAT models with CFSR data substantially improves model performance over forcing the model to use data acquired from local weather stations (please see Saleh et al. (2000) for a case study leading to a similar statement). A clear improvement is obtained using the Tarawneh (2007) produced soil map in preference to the global Waterbase map (e.g. scenario pairs 16/13 and 5/2). As with other statistical measures, inclusion of the Madaba rainfall gauge cause a sharp drop in NSE and increase in RSR for otherwise identical scenarios (e.g. scenario pairs 18/17 and 15/14). Consideration of the three land-use classes
shows a clear variation in uncalibrated model performance, as the global land-use layer out-performs the two locally processed maps.

        Similarly, Table 2 shows the order of NSE and RSR calculated to assess sediment yield prediction. The scenario rank order differs from that in Table 1 due to the different sensitivity of discharge and sediment simulation to various types

of inputs. The global soil map produces considerably poorer model performance than the Tarawneh (2007) map, (e.g. scenario pairs 16/13, 18/15 and 5/2). The comparisons lead to initial classification of scenarios into two groups defined by the specification of the soil input dataset. Within each group the ranking order of land-use/weather combinations is similar. This confirms the high sensitivity of the sediment simulation to input soil data. Across the rest of scenarios, using the CFSR

data results in significantly higher NSE and lower RSR, with a wide gap between them and the successive values (scenarios 18, 12 and downward). This is consistent with the results of the discharge assessment (Table 1). The importance of land-use in determining sediment yield is clear by the priority it takes over the weather in the ranking of scenarios 18, 12, 17, 11, 6, and 5. In all cases, excluding the Madaba rain gauge always yields a closer correlation to observations between scenarios of similar conditions.

**4.2 Case study results**

*4.2.1 Soil data*

The choice of soil dataset is a strong control on model behaviour by all measures (Table 1, Table 2). The pre-calibration performance of the model against both discharge and sediment data is better using the more detailed local soil map (Tarawneh, 2007 and Ministry of Agriculture, 1994) and the CFSR dataset, in combination with the global land-use map.

Conversely, the weakest uncalibrated performance against observations results from applying the global soil map (Waterbase, 2012) and local weather data including the Madaba rain gauge (i.e. the combination of measured and SWAT-generated weather data). It is clear from Fig. 4 that there is a significant difference in the granularity of data between the two input soil maps. The additional detail embodied in the Tarawneh (2007) dataset yields significantly more range in soil class and key SWAT parameters, such as permeability, which presumably directly influences model calculation of both discharge

and sediment loading.

*4.2.2 Weather data*

In contrast to the soil datasets (where more granular, sampled-derived data yield best model performance), the global reanalysis (CFSR) weather data consistently yield better pre-calibration model performance (scenarios 16, 10, 4, 13, 7, 1) than scenarios using locally-recorded weather data (Table 1 and 2). This difference is further exacerbated when the data from

the Madaba recording station is included in the locally-recorded input dataset. A qualitative inspection of rainfall data series shows high values of rainfall recorded at Madaba compared to the global CFSR for similar periods. This in turn influences the extensive infill values generated by the SWAT weather generator for this dataset. Fuka et al. (2014) suggests that using CFSR data provides a remedy to the potential uncertainty linked to using local weather records, which are seldom complete and may not realistically represent the watershed and provide point rainfall measurements neglecting the effects of hydro-

climatic gradients (Ciach, 2003). To understand how prediction of discharge differs between scenarios and for visual comparison between observed and simulated discharges, six scenarios are selected (Fig. 9) to visually assess model

performance. Figure 9 clearly illustrates that better fit is associated with scenarios of lower RSR (closer to zero) and higher NSE and $r^2$ values, with over-prediction resulting from using local weather data regardless of the inclusion of the strongly discontinuous Madaba dataset. Graphical comparison of four sets of observed and simulated sediment yield is displayed in Fig. 10 to demonstrate the tendency of the poorly performing scenarios to significantly overestimate sediment yield. This is

consistent with records containing anomalously high rainfall readings.

### 4.2.3 Land-use data and sensitivity analysis

The only difference between the three scenarios (16, 10, 4) achieving best performance for both discharge and sediment prediction is the land-use data source. These scenarios show good correlation between simulated and observed variables, with lowest $r^2$ and highest NSE and $r^2$. A similar order of the three land-use scenarios is identified for both discharge and

sediment, but the performance of all three scenarios is almost equal. A standard SWAT model 32-parameter global sensitivity analysis (Dechmi et al., 2012; Van Griensven, 2005) is applied using the best-performing scenario 16 to identify quantitatively which internal parameters are the most sensitive for the Wala catchment SWAT model. Table 3 shows the results of this sensitivity analysis using observed discharge and observation-derived sediment load values at the Wala flow station during the simulation period. After discounting parameters which score low sensitivities, it is clear that the seven

highest-ranked parameters are closely related to the properties defined by the soils and land use data inputs.

The most sensitive parameter is the SCS Curve Number (CN), derived directly from land-use data (Neitsch et al., 2011). Nevertheless, our pre-calibration results show that selection among the land-use datasets available in this study yields least influence on model performance. This apparent contradiction can be resolved by inspection of Fig. 3 which shows, in contrast to the soils datasets shown in Fig. 4, that there is relatively little variation both in spatial distribution and range of

physical characteristics among the three available land use maps. Reviewing the CN values for the dominant land-use classes in the three land-use maps, we find them to be close (ranging from 80 to 84) due to the similarity of properties defined for each dataset. Furthermore, the method of HRU definition within SWAT selects the major land-use in each HRU, thus potentially nullifying the gains of higher-resolution land-use maps with numerous smaller land-use classes. While the sensitivity analysis emphasizes the general importance of land-use definition in SWAT catchment models, this case study

shows the value of quantitative interrogation of the available datasets for any specific application.

### 4.2.4 Global reanalysis vs locally-derived datasets

For prediction of discharge (Table 1, Fig. 9), the scenario analysis strongly confirms that the most sensitive constituent is the input weather data. It is obvious to say that precipitation is a fundamental driver of runoff and discharge time-series. However, considerably higher model performance is achieved by the reanalysed CFSR versus local weather datasets,

regardless of the other input datasets. We suggest the reason for this is the continuity and consistency of the CFSR dataset, which is provided by the NCEP reanalysis climate data derived from global satellite imagery for a grid of statistically interpolated points (Saha et al., 2010). Although the local dataset might be expected to capture average daily events more

precisely, this relies on well-calibrated, well-maintained instrumentation and proper representativeness of measurement stations within the study area.

In the Wala catchment, as in many locations world-wide, poor data continuity and reliability necessitate generation of infill data points by the SWAT weather generator. The potential of individual recording stations as a source of error in model output is further demonstrated by the observation that for otherwise similar scenarios, incorporating the Madaba rain gauge (which depends on the SWAT weather simulator to generate 32.75% of its daily records) significantly reduces the performance of the model. This does not fully negate the potential inappropriateness of the weather stations utilized to construct the weather generator, however, by reviewing the statistics generated using these stations, they do not seem to vary significantly from measured weather parameters within the area. We suggest that basic weaknesses in the recording of data are compounded by the challenges posed to the rainfall generator algorithm by strong daily, monthly, and interannual variability in an arid-climate rainfall regime. One possible area for investigation in this respect is the use in the generator of the Markov chain model, which does not account for the interannual variability in the daily weather causing clear inconsistencies with measurements (Jiang et al., 2011). This is crucial in semi-arid and arid regions where precipitation is much more variable on all timescales than in temperate and humid regions.

The control of the choice of soil dataset on model performance is substantial in our analysis, which corresponds with the sensitivity of the model to its internal soil characteristic parameters (Table 3). The results show clear improvement in pre-calibration model performance using higher resolution maps built using field sampling rather than the global map, which is of lower classification quality and resolution, relying heavily on satellite remote sensing. The local soil map yields better pre-calibration performance than the global map, even with different weather data (with/without Madaba gauge), emphasizing the primary importance of soil data in this model of the Wala catchment. Sediment simulation is highly influenced by changes in soil definition; this is expected because soil parameters are directly needed by the USLE (Wischmeier and Smith, 1965) and MUSLE to predict soil erosion and sediment yield and are also required to simulate discharge, which is important for sediment yield prediction.

### 4.3 Effect on calibrated model performance

While it is clear that use of SWAT pre-calibration enables rapid, quantitative comparison of different input datasets, we wished to confirm that optimization of the model in this way yields improved performance after calibration also. To test the functionality of the presented pre-calibration approach in enhancing subsequent calibrated model performance, automatic internal parameter calibration (Abbaspour et al., 2007) is performed for the best (16) and poorest (2) pre-calibration scenarios as described above. Standard SWAT calibration is undertaken by applying consistent conditions and criteria for each scenario separately for discharge simulation. The calibration targets the set of parameters defined in the sensitivity analysis as being the strongest controls on model performance (Table 3). NSE is selected as an objective function and 1000 iterations run.

Table 4 displays a comparison between uncalibrated and calibrated scenarios. Calibration improves the NSE for discharge simulation from 0.56 to 0.64 and from -12 to -11.29 for scenario 16 and scenario 2, respectively. This represents a 14 % performance gain for scenario 16 and a 6 % improvement in scenario 2. It is clear that our pre-calibration methodology accurately reflects the fully-calibrated performance of models based on different input data combinations, yet with a fraction of the computational effort and time. These findings emphasize the value of reducing model uncertainty by undertaking preliminary screening of input datasets and selecting the best available conditions to construct models that achieve the best possible calibrated performance.

## 5 CONCLUSIONS

Previous use of a SWAT model to simulate discharge and sediment yield across the Wala catchment led to a detailed understanding of the hydrological system of the area and the interaction between its components and processes (Ijam and Tarawneh, 2012). In this paper we have developed a discrete methodology (Fig. 11) for using a SWAT model framework, comprising an analytical stage prior to full model calibration, to support decision-making in the selection and application of input datasets for use with catchment hydrological models. This should be of value in the specification and design of catchment modelling in many semi-arid, arid and data-poor regions, since the factorial scenario-testing approach allows rapid, quantitative comparison among a range of datasets of uncertain quality. Model sensitivity to various types and resolutions of data is clear and demonstrates the significant influence of input selection on model performance prior to the calibration step and hence the potential to minimise the computational effort and possible systematic biases inherent in the calibration process (Beven, 2011; Wheater et al, 2008a). In summary, we find:

- continuity and quality of record are critical factors in selecting weather data, over and above use of local measurements. In our case study, inclusion/exclusion of the poor quality, incomplete Madaba dataset results in significant variability in model performance and leads us to recommend the preferential use of global reanalysis data where there is any doubt about local data quality, even in rainfall regimes which are characterised by infrequent, irregular, intense storm events;

- high resolution, high quality soil data (likely to be available only through detailed local survey) yields significant improvements in pre-calibration model performance over globally-available datasets obtained from e.g. remote sensing;

- land-use definition in this specific case at the Wala shows the least impact of the three inputs assessed. We propose this is due to broad similarities between available land use datasets, which is likely to be representative of conditions across much of central Jordan and surrounding arid and semi-arid regions. Our results suggest that only significant and spatially-extensive deviation in actual land use from global freely-available datasets such as GLCC – either as a result

of rapid (or predicted) land development or land degradation – will significantly impact on overall model function and performance.

The key benefit of this work in the context of the Wala Dam and the management of water resources in Jordan is an improvement in the confidence with which catchment data and models can be used in decision making. This applies both to management of existing artificial recharge catchments, such as Wadi Wala, and to the options assessment and selection of new schemes which are critical to securing a more sustainable water resource for the country (JNFP, 2012). The potential utility of SWAT in this context has been previously demonstrated (Ijam and Tarawneh, 2011); this current work provides a rational basis for supporting the selection and use of available input datasets, and targeting of field resources to improve the reliability and coverage of these data.

A general observation is that globally-available weather and land-use datasets tend to perform equal to or better than local data as inputs to the catchment model over a range of dataset combinations, suggesting that these may be preferable sources of inputs where local data is sparse or unreliable. However, we found obtaining a high-quality, ground-truthed soil dataset offers substantial improvements in pre-calibration model performance over regional or global soil datasets. We therefore recommend detailed soil mapping as a priority for targeting desk and field resources to support studies in settings comparable to that studied here. It is highly recommended also to quantitatively and qualitatively improve field measurements to provide trustworthy observations which can be used in model assessment and calibration, for example by increasing the number of gauges within the area, improving the temporal resolution of measurements to involve events at finer intervals, thereby avoiding problems associated with aggregating to coarser intervals. The latter issue is of particular importance in arid and semi-arid regions were hydrologic events are commonly characterized by high intensities and short intervals (daily/sub-daily), therefore, underestimation or misrepresentation of these events may happen when aggregated at coarser time-steps (monthly/yearly). However, feasibility of qualitative and quantitative improvement of data (including input data and observations for model evaluation) should be taken into consideration in order to target important features and optimize cost, time and effort of modelling studies (Hughes, 1995).

## AUTHOR CONTRIBUTION

Tarawneh undertook the SWAT analysis and led the paper writing. Bridge and Macdonald as supervisors contributed to structure and research design and supported the writing of the paper and analysis.

## DATA AVAILABILITY

This work forms part of a thesis at the University of Liverpool which will be submitted in early 2017. At that time all digital data products specifically associated with the work will be made publically available via the University of Liverpool data catalogue (datacat.liverpool.ac.uk). Before then, all enquiries for data can be made to the corresponding author.

**ACKNOWLEDGMENT**

This work is supported by the Mu'tah University under Grant [104/13/30]. The authors would like to recognise the support of Mu'tah University, the Ministry of Water and Irrigation of Jordan and the Wala Dam management in the development of this work.

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

**TABLES**

| Scenario No. | Land-use map[1] | Soil map[2] | Weather data | Madaba station | No. of HRUs | NSE | RSR |
|---|---|---|---|---|---|---|---|
| 16 | c | b | CFSR | – | 47 | 0.56 | 0.66 |
| 10 | b | b | | | 67 | 0.56 | 0.67 |
| 4 | a | b | | | 68 | 0.55 | 0.67 |
| 13 | c | a | | | 48 | -0.32 | 1.15 |
| 7 | b | a | | | 63 | -0.36 | 1.17 |
| 1 | a | a | | | 60 | -0.36 | 1.17 |
| 18 | c | b | Local | Excluded | 47 | -0.36 | 1.17 |
| 12 | b | b | | | 67 | -0.43 | 1.19 |
| 6 | a | b | | | 68 | -0.69 | 1.30 |
| 17 | c | b | Local | Included | 47 | -2.90 | 1.97 |
| 11 | b | b | | | 67 | -3.16 | 2.04 |
| 5 | a | b | | | 68 | -3.56 | 2.13 |
| 15 | c | a | Local | Excluded | 48 | -4.69 | 2.38 |
| 9 | b | a | | | 63 | -4.84 | 2.42 |
| 3 | a | a | | | 60 | -5.39 | 2.53 |
| 14 | c | a | Local | Included | 48 | -11.25 | 3.50 |
| 8 | b | a | | | 63 | -11.42 | 3.52 |
| 2 | a | a | | | 60 | -12.00 | 3.61 |

**Table 1. Number of HRUs and values of NSE and RSR calculated for 18 scenarios for comparison of observed and simulated average monthly discharge ($m^3s^{-1}$) at the Wala catchment outlet. 1 Land-use maps: a) Tarawneh (2007); b) Al-Bakri et al. (2013); c) WaterBase (2012). 2 Soil maps: a) WaterBase (2012); b) Tarawneh (2007).**

| Scenario No. | Land-use map[1] | Soil map[2] | Weather data | Madaba station | No. of HRUs | NSE | RSR |
|---|---|---|---|---|---|---|---|
| 16 | c | b | | | 47 | 0.79 | 0.46 |
| 10 | b | | CFSR | _ | 67 | 0.66 | 0.58 |
| 4 | a | | | | 68 | 0.60 | 0.64 |
| 18 | c | | Local | Excluded | 47 | -0.11 | 1.06 |
| 12 | b | | | | 67 | -0.11 | 1.06 |
| 17 | c | | Local | Included | 47 | -1.67 | 1.63 |
| 11 | b | | | | 67 | -1.81 | 1.68 |
| 6 | a | | Local | Excluded | 68 | -2.97 | 1.99 |
| 5 | a | | Local | Included | 68 | -7.21 | 2.86 |
| 13 | c | a | | | 48 | -12.74 | 3.71 |
| 7 | b | | CFSR | _ | 63 | -16.47 | 4.18 |
| 1 | a | | | | 60 | -22.70 | 4.87 |
| 15 | c | | Local | Excluded | 48 | -26.72 | 5.26 |
| 9 | b | | | | 63 | -36.01 | 6.08 |
| 14 | c | | Local | Included | 48 | -42.16 | 6.57 |
| 8 | b | | | | 63 | -48.98 | 7.07 |
| 3 | a | | Local | Excluded | 60 | -59.72 | 7.79 |
| 2 | a | | Local | Included | 60 | -85.06 | 9.28 |

**Table 2. Number of HRUs and values of, NSE and RSR calculated for 18 scenarios for comparison of observed and simulated average monthly sediment yield (t/month) at the Wala catchment outlet. 1 Land-use maps: a) Tarawneh (2007); b) Al-Bakri et al. (2013); c) WaterBase (2012). 2 Soil maps: a) WaterBase (2012); b) Tarawneh (2007).**

| Name | Description[1] | Rank |
|------|----------------|------|
| CN2 | Initial SCS CN II value (Curve Number) | 1 |
| SOL_AWC | Available water capacity (mm $H_2O$/mm soil) | 2 |
| SOL_Z | Soil depth (mm) | 3 |
| SURLAG | Surface runoff lag time (days) | 4 |
| ESCO | Soil evaporation compensation factor | 5 |
| CH_N | Manning's n value for main channel | 6 |
| ALPHA_BF | Baseflow alpha factor [days] | 7 |

[1] (Van Griensven, 2005)

**Table 3. Results of parameters sensitivity analysis of scenario 16**

| Scenario No. | NSE (uncalibrated) | NSE (calibrated) |
|:---:|:---:|:---:|
| 16 | 0.56 | 0.64 |
| 2 | -12.00 | -11.29 |

**Table 4. Values of NSE calculated for uncalibrated and calibrated best and poorest-performing scenarios (16 and 2, respectively) for discharge simulation.**

**FIGURES**

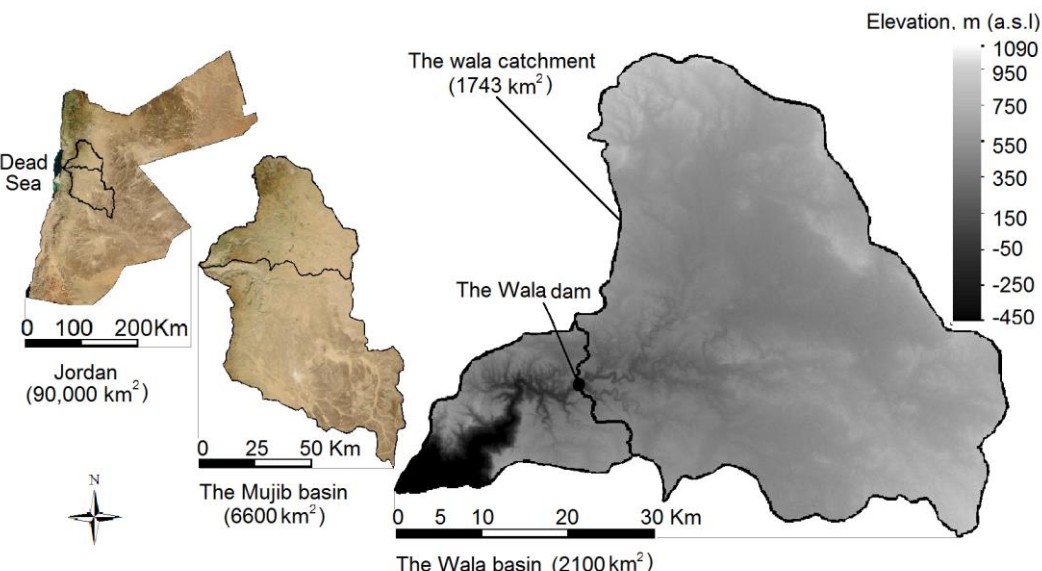

Figure 1. Location of the Wala Catchment.

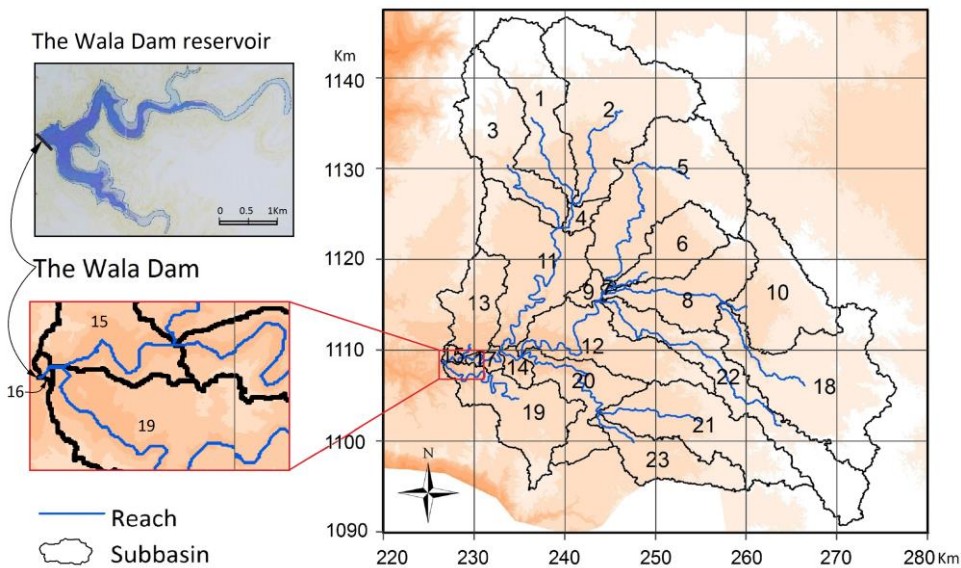

**Figure 2. The Wala catchment delineation into subbasins, stream pattern and the catchment outlet.**

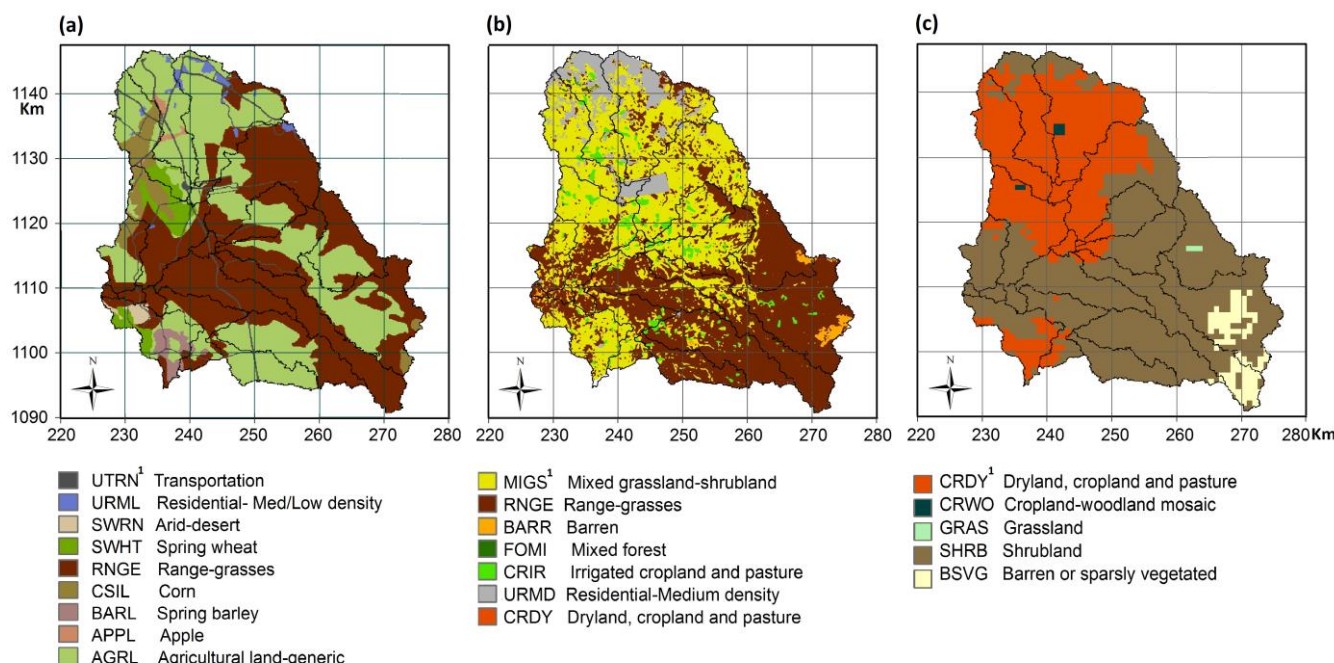

**Figure 3. Land-use classification over the Wala catchment: a) Tarawneh (2007); b) Al-Bakri et al. (2013) ; c) WaterBase (2012). 1 SWAT land-use codes (Arnold et al., 2013).**

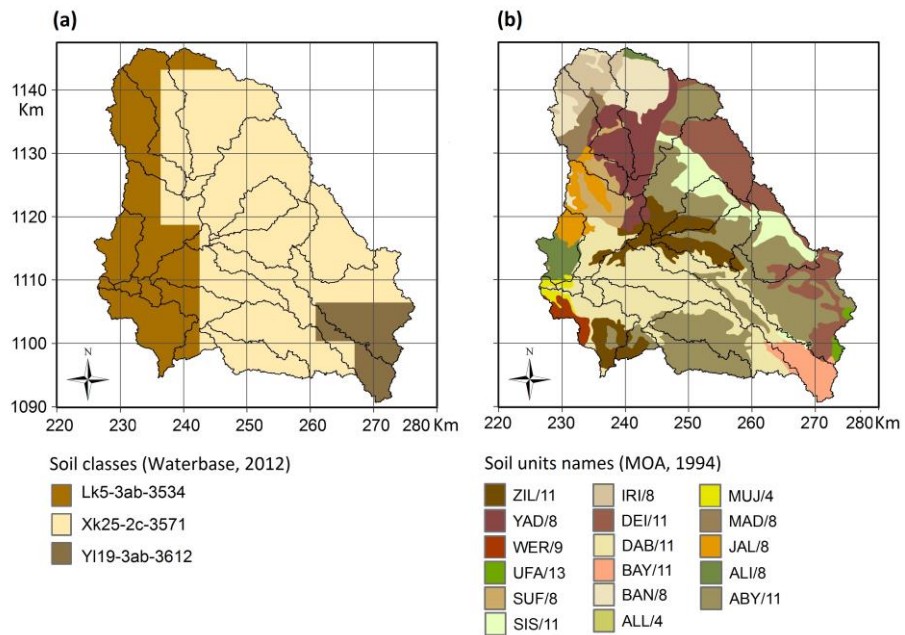

**Figure 4. Soil classification of the Wala catchment: a) WaterBase (2012); b) Tarawneh (2007).**

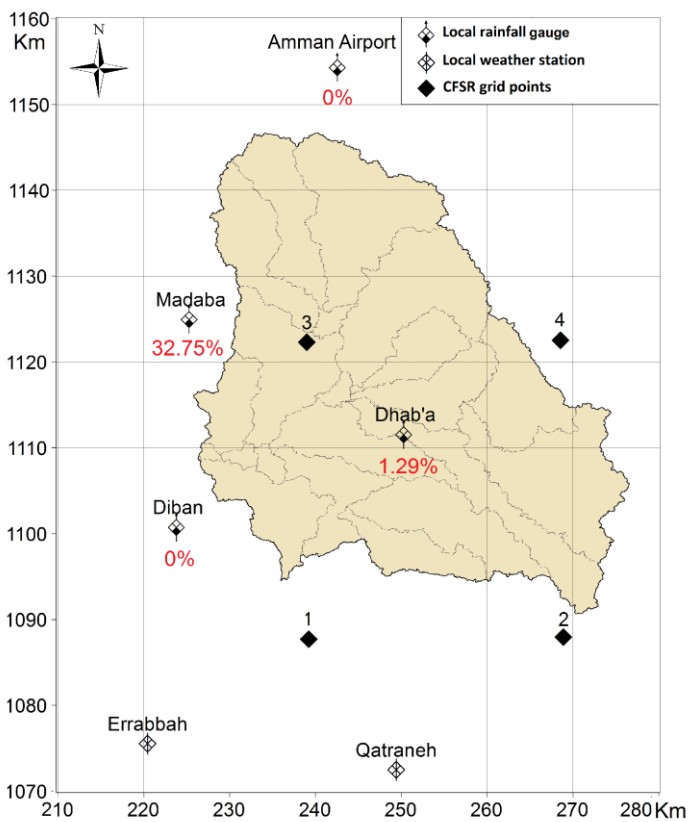

**Figure 5. Local weather stations, CFSR grid points; and local rainfall gauges with their percentage of missing data over the period 1971-2002.**

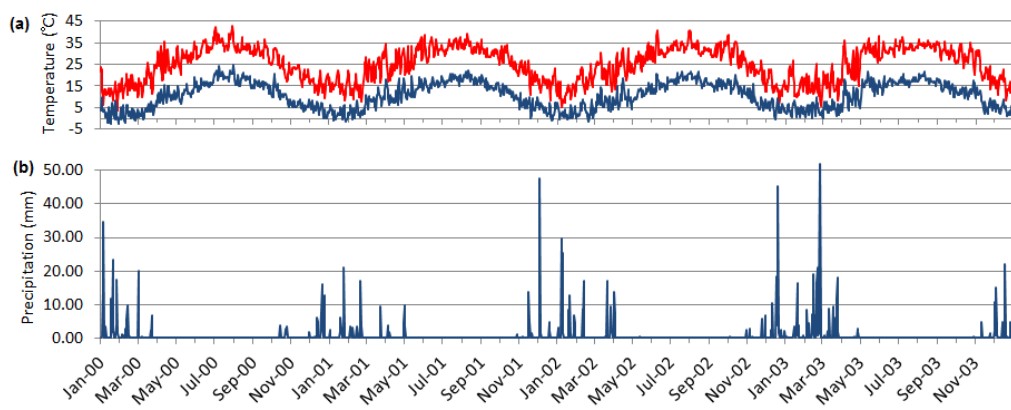

**Figure 6. (a) Daily minimum and maximum temperature of Qatraneh station; (b) Daily precipitation of Amman Airport gauge, for the period 2000 - 2003.**

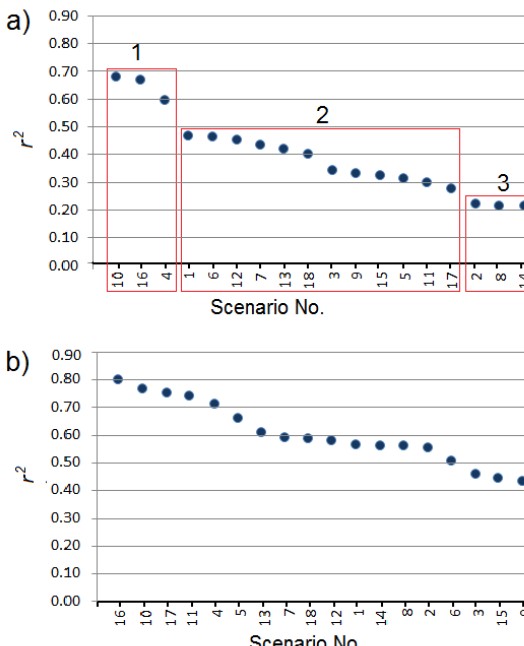

**Figure 7. Values of *r²* calculated for prediction of a) discharge; b) sediment yield, from the 18 scenarios.**

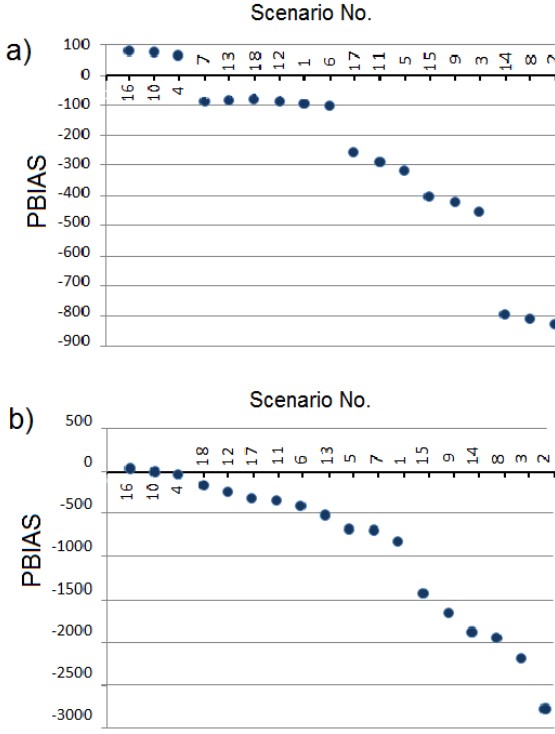

**Figure 8. Values of PBIAS calculated for prediction of a) discharge; b) sediment yield, from the 18 scenarios.**

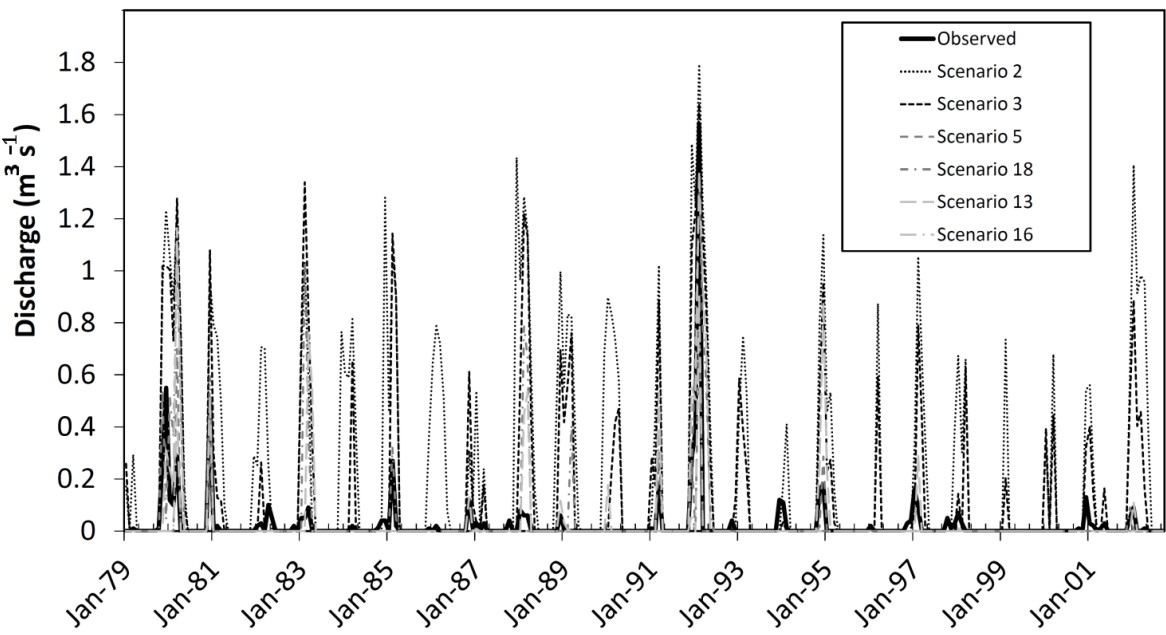

**Figure 9. Simulated and observed average monthly discharge (m³s⁻¹) for scenarios 2, 3, 5, 18, 13 and 16.**

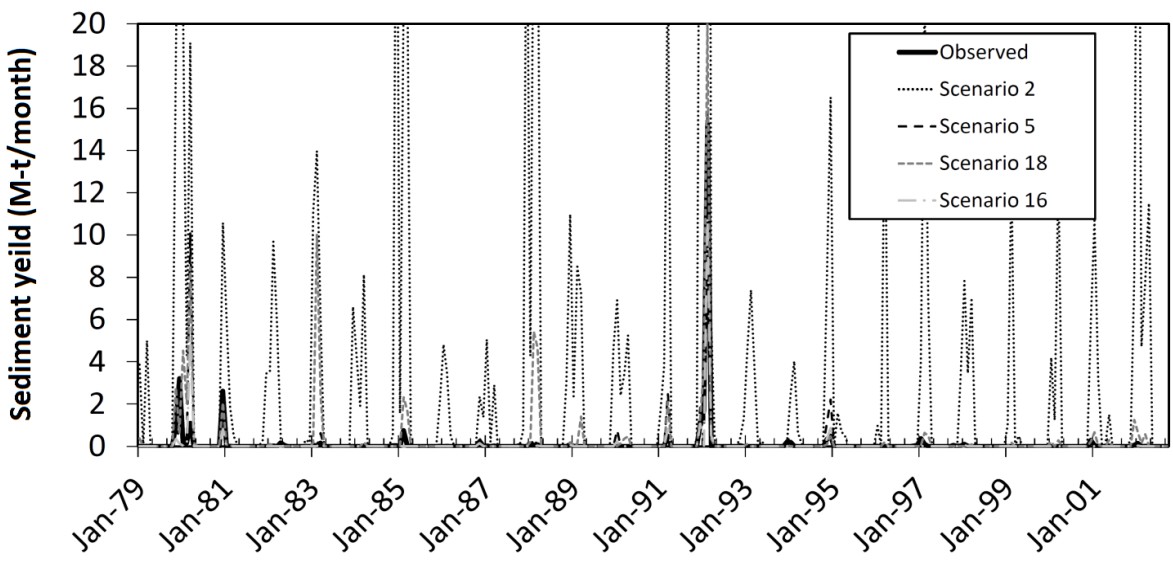

**Figure 10. Simulated and observed monthly sediment yield (M-t/month) for scenarios 2, 5, 18 and 16.**

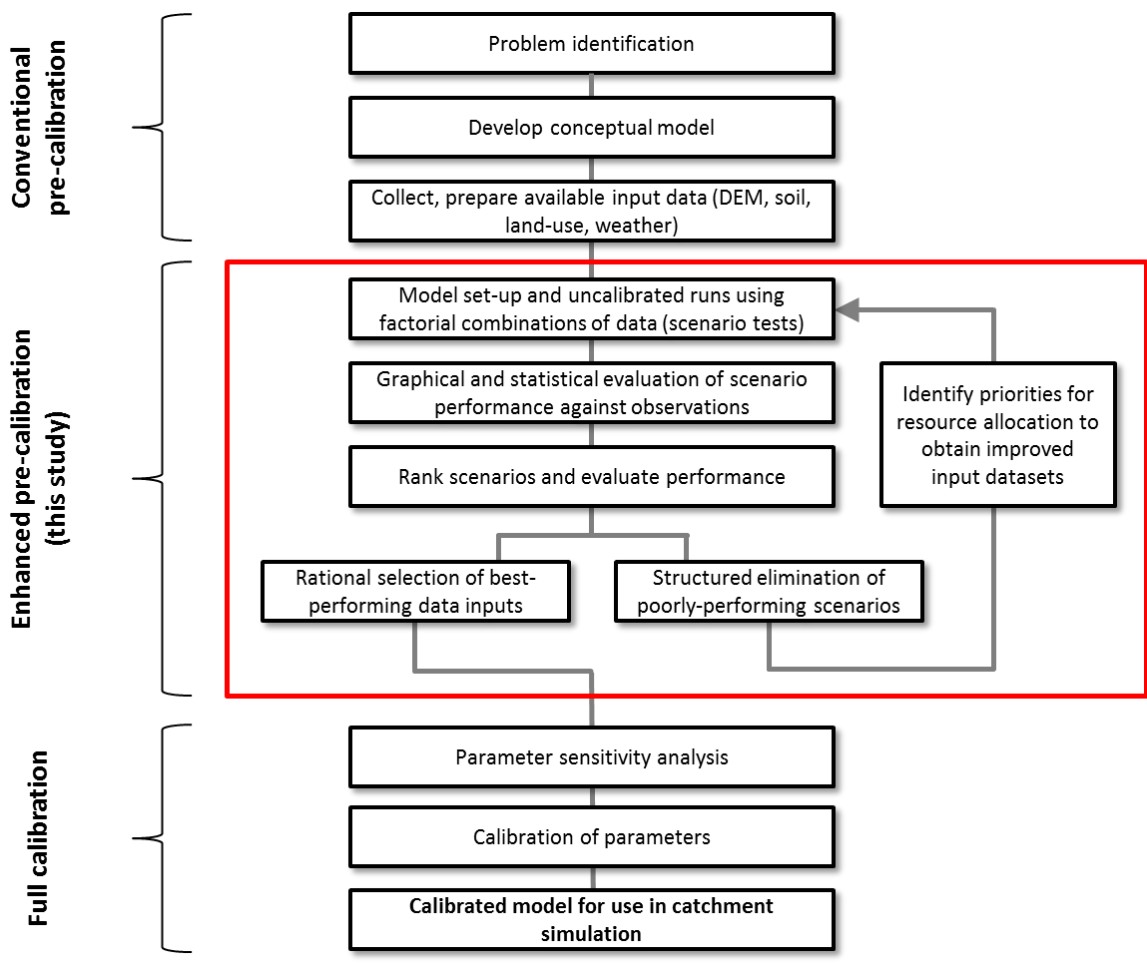

**Figure 11. Workflow illustrating a generic pre-calibration approach based on the methodology outlined in this study.**