# Peer review of "A pre-calibration approach to select optimum inputs for hydrological models in data-scarce regions"

_Hydrology and Earth System Sciences, 2016_

## Referee Comment (RC1) · Anonymous Referee #1 · 27 Jun 2016

**1. General comments**

The authors addressed a major issue of dealing with scarce data having different sources as well temporal and spatial scales for hydrological modelling purposes.

The methodology was very clear and the obtained results are of high importance particularly for hydrologists and soil and water conservation specialists working in dry environments.

The paper was very well written and illustrated.

The bibliography is complete and relatively up to dated.

**2. Specific comments**

• It is recommended to further justify the choose of the SWAT model (P4) with case applications particularly in similar environments, • For the land use maps (p5): o Did you consider that there were no major changes during the simulation period ? o A part from the Wala dam, are there other hydraulic structures (soil and water conservation, water harvesting, etc.) in the watershed. If yes, how did you represent them in the model ? • Provide a summary of the used soil characteristics for the two soil maps (p5-6) and indicate properly the measured and the estimated ones • Using average monthly discharge data (p7) in a dry environment needs to be well explained and justified. In fact, in these areas, flood events occur generally in most of the cases in very short periods (some hours). Therefore, even using daily averages may cause some problems with model calibration and validation !. • What do you mean by "Howard Humphreys and Partners (1992) identify a strong log linear relationship . . . . . ... after Tarawneh (2007)" ? (p7). May I understand that the sediment yield was estimated based on this relationship ?. • In Figures 3 (P24) and 4 (P25): are all these classes exist in the study watershed ?

**3. Technical corrections**

* P5 L10: Replace 'Luzio et al., 2002' by 'Di Luzio et al., 2002' (it is the same author) * P5 L26: Check if it is Leon, 2007 or Leon, 2013 * P6 L23: Replace 'by (Neithsch et al., 2001)' by 'by Neithsch et al. (2001)' * P7 L10: Replace 'see for exampleZhang' by 'see for example Zhang' * Ageena et al (2014): not found in the text * Ageena et al. (2013): not found in the text * P16 L36: Check if the reference of Montheith is complete ? * P17 L22: Check if you need to type twice 2009 * P17 L34: Check if you need to type twice 2008b * P27: Correct 2000-200 !! * Figure 9 (P30): Check if the scenarios 13, 18, 5 and 3 are included in the graphs !

---

## Referee Comment (RC2) · Anonymous Referee #2 · 7 Jul 2016

**A: GENERAL COMMENTS**

The manuscript investigates sensitivity of the SWAT model in Wala catchment in Jordan. The main idea of the article is to develop a framework to test the effects of various data sets in hydrological models to support water management and planning in data scarce regions. The results also support to identify gaps that need to be filled by e.g. improved monitoring. For this purpose the authors tested extends of errors in predictions due to the use of different types of input data to SWAT. They developed eighteen hydrological models (using combination of three land use maps, two soil maps, and three climate time series from the local and global sources), and they evaluated the models using measured monthly discharge and constructed sediment yield data at the

outlet of the watershed. The authors showed that significant performance gain can be obtained with the proper combination of inputs. They conclude that selection of quality data will reduce uncertainty of hydrological model outputs.

The subject falls within the general scope of the journal. The aims of the study are interesting for the readers of the HESS. The obtained results appear encouraging. It is however strongly advisable to extend the following points in the paper:

1. This study tests relative quality of the existing datasets from local/global sources to support the statement in "Page 2, L22-29" on reducing uncertainty in data poor regions when transferring parameters/knowledge from neighboring or geographically similar catchments. While the general idea is very interesting, but I felt the main hypothesis never tested. I think the authors can improve novelty of the work by quantifying how the traditional knowledge (parameter) transfer from the neighboring watersheds (e.g., using calibrated parameters of poor model-scenario in this study) versus parameter transfer from a better model (using quality data model-scenario in the study) help reduce uncertainty in model predictions of the data poor regions. One would expect that this could be done by classical sample test in small portion of the watershed.

B. SPECIFIC COMMENTS

1. Authors may provide more background about previous SWAT applications in the study region (if published in any peer reviewed journals or reports that are available for public); and also strength and limitation of the model in behavior simulation of the major hydrological events in such arid environment with intense, highly intermittent, and often localized storms. Authors may add this in the "model selection" and "discussion" parts. Authors emphasized on the importance of input data uncertainty but never discussed other sources of errors in hydrological modeling: e.g., model structure (process simplifications, which might be case in this work), and parameter estimations.

2. Page 6, L23-24: How the two weather stations (Qatraneh and Errabbah) represent climate conditions in the study area? If the stations are not representing actual conditions, the generated data (to fill the gaps in the recorded time series) will be subjective, and as a result poor hydrological performance will be obtained (it is seen also from the results).

3. Page 6, Section 3.5: How the HRUs were defined? Please indicate if you used dominant or multiple (/threshold?).

4. Page 7, Section 3.7: In this study the sediment yields are not measured but estimated using the streamflow data. This cause a subjective comparison results when testing different scenarios in this study: (i) It is obvious from the results that model-scenarios that perform better in the simulation of streamflow, present higher performance in modeling sediment too. Therefore, any judgment on the performance of the input data in model simulation will be subjective. (ii) The inherent errors in the estimated sediment yields may be compromised/offset by the model prediction errors due to less quality input data, resulting in a wrong conclusion in scenario selection. The authors may provide a background on how the sediment data were estimated and how the above mentioned points may be justified when evaluating model-scenarios in this work.

5. Page 7, L 29-31: The strong relationship between Q and T should be explained. The provided reference is not enough and not available for readers.

6. There are many studies suggesting multi gauge evaluation (rather than single outlet comparison) in spatially explicit hydrological modeling, to prevent the spatial errors in the upstream catchments that may offset and not observed in the outlet. If the gauge data are not available in other tributaries of this study to conduct multi-gauge evaluation, at least the importance of this effort should be highlighted to also support the statement on Page 3, L3-5, on the investment efforts to improve the limited data.

7. Authors may add three other layers of data into Fig. 5: (i) river network; (ii) streamflow station (Wadi Wala) where the models were evaluated; (iii) Wala dam. With the geographic coordinates only, it is hard to understand the system.

8. Page 7, L27-28: The authors evaluated their model-scenarios using the discharge data of Wadi Wala station that is located downstream of the dam location. How the operation of dam and the effects on hydrological regime in downstream station was considered? Authors may explain if the operation of dam was simulated in the model.

9. Page 7, L32: The "parameterised" SWAT model is not clear. Does it mean that authors make change in model parameters (for any calibration purposes?). If so, it contradicts with several statements in the text that that the evaluations were performed prior to calibration.

10. Page 7, L32: SWAT model operates on "daily" not monthly or yearly or seasonal. Please rephrase the sentence. Authors may aggregated data from daily to monthly etc.

11. Page 9, L7-8: "NSE drops ... and all cases". These are interesting results. Authors may explain why CFSR data performed better than local observations. Are the findings consistent with other studies around the world that applied CFSR in hydrological models?

12. Page 9, L11-13: usually a better performance is expected when using locally produced high resolution maps (e.g., the one from Al Bakri). Authors may explain the possible reasons why the course resolution map of global source performed better in this study.

13. Page 10, section Weather data: Please refer to my previous comment in this section (comment #2).

14. Page 10, L12-16: (i) When looking at the top left graph in Fig. 9, the high performing statistics are usually representing low flows, while high flows are almost completely underestimated (or not predicted at all). How the authors will consider both low flows (important for drought) and high flows (major events and important for flooding and water saving) explain in their model evaluation and scenario selection? (ii) As indicated in several places in the text, the Wala basin is located in semi-arid area that is prone

to intense (sub-daily) rainfall events. My concern is that how model evaluation at the monthly scale will ensure representation of locally important short-term events? (iii) As mentioned previously, the data are compared at downstream station that might be affected by operation of a dam.

15. Page 11, L13-22: Another reason is the use of statistics generated from inappropriate weather stations: please refer to my comment in this section (comment#2)

16. Page 11, section 4.3: It would be interesting to apply the calibrated parameters of the model-scenarios to support the main idea of the study on how appropriate model setup and quality data help managers of data scarce watersheds when transferring knowledge (e.g., parameters) from neighboring watersheds. Please refer to my general comment.

C. TECHNICAL CORRECTIONS

1. Authors may keep the acronyms consistent: either provide full definition for all of the NSE, RSR, and PBIANS; with the acronyms in the bracket; or provide acronyms only. 2. Page 3, L3: "..quality and/or quality" : "..quality and/or quantity". 3. Page 2, L6: "..with extremely . . . and" is repetition of previous statement. Either remove this sentence or rephrase. 4. Page 4, L16: "basin characteristics" are NOT represented by the USDA CN method, but the "surface runoff' is simulated. 5. Page 24, Fig. 5: for the ease of visual comparison, I suggest using similar coloring pattern for the same LU classes in the three maps. 6. Page 6, section 3.4.3: the authors may move this short paragraph to section 3.3, where you first discuss the DEM in the study area. 7. Page 7, L23: "Running SWAT . . .for full description": statement does not fit in this section. I suggest to move it to the section 3.2, where the authors introduce SWAT model, or to remove it. 8. Page 30, Fig. 9: Titles and legends are too small. Only scenario 16, and scenario 2 are presented in the figure, while the caption indicates scenarios 13, 18, 5, and 3 as well as 16, and 2. Please carefully check the figures, captions, and discussion in the text to match these three parts. 9. Page 30, Fig. 9: the explanation of

the sediment yield graphs are missing from the caption and text. 10. Page 31, Fig. 10: please provide high quality graphs with consistent size and formatting, and consistent scale in the vertical axis to help comparison of the graphs. With the current format it is hard to read and compare the graphs.

---

## Author Comment (AC1) · 22 Jul 2016

The authors would like to express appreciation to Referee #1 for the time and effort dedicated to review the manuscript and provide in-depth comments, important suggestions and accurate technical corrections, which have considerably improved the quality of the manuscript. We would also like to thank the Referee for finding the potential of the presented work to assist areas most in need for such approaches and for positively commenting on the results, methodology, structure and technical status of the manuscript. We wish our point-by-point response addresses the Referee's concerns and improves the manuscript.

Kindly, please find attached to this comment a supplementary zip file containing three

pdf files: 1. The authors' point-by-point response to Referee #1. 2. Revised version of the manuscript. 3. Revised version of the manuscript in 'track-changes' style to facilitate finding modifications.

Please also note the supplement to this comment:
http://www.hydrol-earth-syst-sci-discuss.net/hess-2016-242/hess-2016-242-AC1-supplement.zip

---

## Author Comment (AC2) · 22 Jul 2016

The authors would like to express thankfulness to Referee #2 for the thorough review, thoughtful suggestions and raising critical points for discussion. It is a pleasure that this valued review finds the aims of our study interesting and the results encouraging. We have taken all comments and suggestions of Referee #2 on board and prepared a detailed point-by-point response with applying the required modifications across the manuscript. We hope our response clears up the Referee's concern and strengthens our work.

Kindly, please find attached to this comment a supplementary zip file containing three pdf files: 1. The authors' point-by-point response to Referee #2. 2. Revised version

of the manuscript. 3. Revised version of the manuscript in 'track-changes' style to facilitate finding modifications.

Please also note the supplement to this comment:
http://www.hydrol-earth-syst-sci-discuss.net/hess-2016-242/hess-2016-242-AC2-supplement.zip

---

## Author Comment (AC3) · 2 Aug 2016

In response to the Referees' comments regarding the system and features of the Wala catchment and Dam, we would like to add the attached supplementary .zip file containing: 1. Wala_chatchment.mp4: Video showing the study area (the Wala catchment, Jordan).

2. Reach.kmz: main reaches of the Wala catchment (Please view in Google Earth)

3. Subbasin.kmz: catchment delineation into 23 subbabsins (Please view in Google Earth).

We hope these enable better understanding of the catchment system, terrain, drainage

pattern, location of the dam, broader context of the region and how the model is of relevance to Jordan.

Please also note the supplement to this comment:
http://www.hydrol-earth-syst-sci-discuss.net/hess-2016-242/hess-2016-242-AC3-supplement.zip

———————————————————

---

## Author Response (AR1)

**Ms Esra'a Tarawneh**

School of Engineering
The Quadrangle
The University of Liverpool
Brownlow Hill
Liverpool L69 3GH

E erat@liverpool.ac.uk

www.liv.ac.uk

30th August 2016

Dear Editor,

**Re. Resubmission of manuscript HESS-2016-242**

We are pleased to resubmit for publication the revised version of manuscript HESS-2016-242 "*A pre-calibration approach to select optimum inputs for hydrological models in data-scarce regions.*" We appreciate the positive opinion and constructive criticisms of the Editor and the Referees.

Please find enclosed with this letter a revised manuscript and associated documents (Authors' response to Referees and marker-up manuscript (later in this document) and a supplementary short video of the study area) in respect of our resubmission of the above paper.

We have taken the opportunity to carefully review and revise the manuscript in accordance with the detailed and thoughtful criticisms and suggestions received. We hope that our responses to all the general, specific and technical reviews, both in the point-by-point commentary and on the evidence of the revised manuscript itself, satisfy you and the Referees that the concerns have been addressed resulting in a paper which will be acceptable for publication.

Our justifications for publication in *Hydrology and Earth System Sciences,* assertions of significance and confirmations in respect of authorship and prior publication remain as per our original submission dated 23rd May 2016.

Yours sincerely,

Ms Esra'a Tarawneh

on behalf of co-authors Dr Jonathan Bridge and Dr Neil Macdonald.

**Authors' Response to the Review Comments**

**Editor (Dr Elena Toth) – Received 11th August 2016**

1. **EC:** The paper presents a comparison of the rainfall-runoff modelling results obtained with different input and watershed characterization, applied to a data-scarce region, well representative of many parts of the worlds, and therefore may certainly be of interest for a wide hydrologists' audience. The two referees have a positive opinion of the paper and I warmly invite the Authors to submit the revised version they are already working on, taking into account all the comments made by the Referees.

   **AC:** The Authors would like to thank the Editor for handling the review process and sharing a positive opinion of the paper and constructive comments to improve it. We appreciate the invitation to submit the revised manuscript and hope the improvements we have made by carefully following the review instructions properly address the concerns of the Editor and the Referees.

2. **EC:** I will ask the opinion of Ref 2 on the revised version, while I believe that the Authors have properly addressed the points raised by Ref 1, but the use of monthly instead of daily data (see comments by Ref 1 and the Authors' reply) should be better clarified in the text (it is not explained in the revised version) and consider adding in the paper the results got at daily scale.

   **AC:** We appreciate the importance of this point and hope our response addresses it. Please see the addition at the end of Section 3 and our revised response to specific comment #5 of Referee #1 below.

3. **EC:** On the comparison of rainfall input: the authors themselves acknowledge in the discussion of the results, the use of a rainfall generator is questionable for filling time-series (and even more so in this case, where the two stations are far from the catchment and their data are not even included in the daily local set). But since the performances are very poor also when excluding the only station where the weather generator is massively used, there are certainly problems in the ability of the remaining 3 raingages (where there almost no missing data, so the weather generator is not used) to represent the precipitation field over the catchment, probably due also to the lack of gauges in the eastern part of the study area; the authors should consider further comments on such issue (some comments are actually already included in the revised version).

   **AC:** We fully agree with this point and have tried to clarify it in the revised text. Please see the modified sections 3.6 and 4.1.

**Anonymous Referee #1 - Received and published 27th June 2016**

**A. General comments**

**RC:** The authors addressed a major issue of dealing with scarce data having different sources as well temporal and spatial scales for hydrological modelling purposes. The methodology was very clear and the obtained results are of high importance particularly for hydrologists and soil and water conservation specialists working in dry environments. The paper was very well written and illustrated. The bibliography is complete and relatively up to dated.

**AC:** The authors would like to express appreciation to Referee #1 for the time and effort dedicated to review the manuscript and provide in-depth comments, important suggestions and accurate technical corrections, which have obviously improved the quality of the manuscript. We would also like to thank the Referee for finding the potential of the presented work to assist areas most in need for such approaches and for positively commenting on the results, methodology, structure and technical status of the manuscript. We wish the point-by-point response below addresses the Referee's concerns and improves the manuscript. A modified version of the manuscript will be uploaded

**B. Specific comments**

1. **RC:** It is recommended to further justify the choose of the SWAT model (P4) with case applications particularly in similar environments:
   **AC:** Section (3.2 Model selection and structure) is modified to address this important suggestion.

2. **RC:** For the maps (p5): Did you consider that there were no major changes during the simulation period?
   **AC:** Yes, from the authors' knowledge of the relatively slow land use change in the catchment supported by the available information/maps, it can be assumed that land use change is negligible for this study. Other studies have found that main changes have happened by 1978 (our study starts from 1979). Please see the modified section (3.4.1 Land use).

3. **RC:** A part from the Wala dam, are there other hydraulic structures (soil and water conservation, water harvesting, etc.) in the watershed. If yes, how did you represent them in the model?
   **AC:** Not up to the authors' knowledge or published work and if any, they can be just small farm ponds/traps. The biggest in the area is the Wala dam with a reservoir surface area of 0.8 Km$^2$ (compared to ~ 2 000 Km$^2$ catchment) and it was constructed at the outlet in 2002 (hence, not represented in the model). A note added within Section 3.7 (Scenario comparison) regarding the beginning of impoundment. Please find the supplementary short video of the Wala Catchment for a better display and understanding of the study area.

4. **RC:** Provide a summary of the used soil characteristics for the two soil maps (p5-6) and indicate properly the measured and the estimated ones.
   **AC:** Comment addressed and section (3.4.2 Soil) is modified accordingly.

5. **RC:** Using average monthly discharge data (p7) in a dry environment needs to be well explained and justified. In fact, in these areas, flood events occur generally in most of the cases in very short periods (some hours). Therefore, even using daily averages may cause some problems with model calibration and validation!

**AC:** We appreciate that ideally the use of daily information as both input and output would be preferable. Unfortunately, attempting to run the analysis at the daily temporal scale failed to achieve satisfactory results, this we feel could be the result of several factors (e.g. catchment lag, partial precipitation coverage of the catchment, data quality issues). The use of the monthly provided a more satisfactory output, we feel that this is the result of the temporally short (sub)daily precipitation being averaged over the month, removing the ephemeral nature of precipitation and subsequent river flows and sediment production. The original discharge measurements were daily and their monthly averages were calculated to match the model output time interval. Please see the edits in paragraph 2 of section 3.7 and the text below as inserted at the end of the section:

"Whilst input data (climatic) are based at a daily temporal scale, the model outputs are considered at a monthly timescale for several reasons, i) daily observations of discharge and sediment are unavailable at the Wala station for the whole period of study, with only monthly observations available for model evaluation; ii) A shorter period (1990 – 1996) of daily observations are available at Wala station, but using these yields poor correlations <0.1 between daily model-simulated and observed discharge, iii) with incomplete/low quality measurements, potential for lag  within the pairs of daily simulated and observed values (for model statistical evaluation) can present challenges, which can be reduced when using aggregated temporal data; and most importantly, iv) the objective of this study is to determine long-term flux within the catchment, avoiding the complexity presented by ephemeral systems and since the monthly comparison achieves reasonable fit between observed and simulated values, it is considered sufficient for evaluation of the current model with more convenience. However, all calculations of the model occur on a daily time step, which ensures that hydrological events are accounted for separately as they occur each day.

Similar approach is adopted in several studies, particularly using SWAT, in both humid and arid regions. Spruill et al. (2000) evaluate daily and monthly SWAT models simulation for a small watershed in central Kentucky and state that SWAT is an efficient tool for monthly runoff simulation with NSE values of 0.58 – 0.89 compared to - 0.04 – 0.19 for daily runoff simulation during the same period. The reason suggested is that the model poorly detects peak flows and recession rates while it performs better with total monthly values. For reasonable performance of SWAT, Huang and Zhang (2004) select to simulate discharge in a semi-arid catchment in China on a monthly basis, which leads to NSE of 0.88. The difference between daily and monthly simulations is investigated in watersheds of different scales by Heathman and Larose (2007) and the results show that simulating higher discharge rates, which is usually associated with larger watersheds, introduces greater uncertainty in SWAT discharge estimates and the study states that very good model performance is achieved for monthly stream-flow estimation while the outputs of daily simulation are only within acceptable range."

6. **RC:** What do you mean by "Howard ´ Humphreys and Partners (1992) identify a strong log linear relationship . . .. . ... after Tarawneh (2007)" ? (p7). May I understand that the sediment yield was estimated based on this relationship?

   **AC:** Yes, there are no direct measurements for the study period but the available sediment measurements (for a different period) are linked to discharge by that relationship for the purposes of designing the dam and managing the catchment. Hence, measured discharge is projected to that linear relationship to derive relevant sediment values (as the best available source of data). Equation added and text edited for clarification (section 3.7 Scenario comparisons).

7. **RC:** In Figures 3 (P24) and 4 (P25): are all these classes exist in the study watershed?

   **AC:** Yes, according to the cited land use maps and general knowledge of the catchment supported by a field visit by the authors in 2013.

**C. Technical comments**

1. **RC:** P5 L10: Replace 'Luzio et al., 2002' by 'Di Luzio et al., 2002' (it is the same author):

   **AC:** Done.

2. **RC:** P5 L26: Check if it is Leon, 2007 or Leon, 2013 :

   **AC:** It is actually 2011 (Version 2). Edited in text and bibliography.

3. **RC:** P6 L23: Replace 'by (Neithsch et al., 2001)' by 'by Neithsch et al. (2001)'

   **AC:** Done.

4. **RC:** P7 L10: Replace 'see for example Zhang' by 'see for example Zhang'

   **AC:** Done

5. **RC:** Ageena et al (2014): not found in the text

   **AC:** EndNote error: Author name was missing and only years appear. Reference added properly.

6. **RC:** Ageena et al. (2013): not found in the text.

   **AC:** EndNote error: Author name was missing and only years appear. Reference added properly.

7. **RC:** P16 L36: Check if the reference of Montheith is complete?

   **AC:** Reference completed.

8. **RC:** P17 L22: Check if you need to type twice 2009

   **AC:** Just once – edited.

9. **RC:** P17 L34: Check if you need to type twice 2008b

   **AC:** Just once – edited.

10. **RC:** P27: Correct 2000-200 !!

    **AC:** Corrected to 2003.

11. **RC:** Figure 9 (P30): Check if the scenarios 13, 18, 5 and 3 are included in the graphs!

    **AC:** Apologies, wrong figure was inserted – corrected now including the scenarios mentioned.

**Anonymous Referee #2 - Received and published 7th July 2016**

**A. General comments**

**RC:** The manuscript investigates sensitivity of the SWAT model in Wala catchment in Jordan. The main idea of the article is to develop a framework to test the effects of various data sets in hydrological models to support water management and planning in data scarce regions. The results also support to identify gaps that need to be filled by e.g. improved monitoring. For this purpose the authors tested extends of errors in predictions due to the use of different types of input data to SWAT. They developed eighteen hydrological models (using combination of three maps, two soil maps, and three climate time series from the local and global sources), and they evaluated the models using measured monthly discharge and constructed sediment yield data at the outlet of the watershed. The authors showed that significant performance gain can be obtained with the proper combination of inputs. They conclude that selection of quality data will reduce uncertainty of hydrological model outputs. The subject falls within the general scope of the journal. The aims of the study are interesting for the readers of the HESS. The obtained results appear encouraging. It is however strongly advisable to extend the following points in the paper:

1. This study tests relative quality of the existing datasets from local/global sources to support the statement in "Page 2, L22-29" on reducing uncertainty in data poor regions when transferring parameters/knowledge from neighboring or geographically similar catchments. While the general idea is very interesting, but I felt the main hypothesis never tested. I think the authors can improve novelty of the work by quantifying how the traditional knowledge (parameter) transfer from the neighboring watersheds (e.g., using calibrated parameters of poor model-scenario in this study) versus parameter transfer from a better model (using quality data model-scenario in the study) help reduce uncertainty in model predictions of the data poor regions. One would expect that this could be done by classical sample test in small portion of the watershed.

**AC:** The authors would like to express thankfulness to Referee #2 for the thorough review, thoughtful suggestions and raising critical points for discussion. It is a pleasure that this valued review finds the aims of our study interesting and the results encouraging. We have taken all comments and suggestions of Referee #2 on board and prepared the following point-by-point response with applying the required modifications across the manuscript. We hope our response clears up the Referee's concern and strengthens our work.

1. We appreciate the Referee's recommendation and apologize for the confusion that made the statement a bit unclear to readers. We would like to clarify that the statement in this paragraph, which states our driving research questions, is in the last two lines: 'which datasets should be employed in modeling and where should investment be targeted to improve data quality?', which we discussed later, for example by recommending improving soil data for their high importance. The lines before that compare between geomorphically similar (e.g. humid) and significantly different (e.g. semi-arid) areas and show that transfer of data/parameters is likely to help in humid areas, but not necessarily applicable in dry lands and this is why we suggest our approach to select suitable data to model dry lands and invest on improving the most sensitive datasets rather than depending on transfer of parameters which is expected to bring uncertainty.

   We do not claim that our approach reduces uncertainty when parameters are transferred from neighbouring areas; rather we encourage testing data available for the area itself and recommend improvement/investment where possible.

We do not currently have parameters from neighboring areas to compare with our poor and good scenarios but this would be a good test to plan for future work. Our main hypothesis is literally stated in the last paragraph of the section.

Please see the modified text; we hope it makes clear our objectives and hypothesis.

**B. Specific comments**

1. **RC:** Authors may provide more background about previous SWAT applications in the study region (if published in any peer reviewed journals or reports that are available for public); and also strength and limitation of the model in behavior simulation of the major hydrological events in such arid environment with intense, highly intermittent, and often localized storms. Authors may add this in the "model selection" and "discussion" parts. Authors emphasized on the importance of input data uncertainty but never discussed other sources of errors in hydrological modeling: e.g., model structure (process simplifications, which might be case in this work), and parameter estimations.

   **AC:** Please see the modified section 3.2 (Model selection and structure) for improved background about SWAT and 1 (Introduction) for further details about uncertainty.

2. **RC:** Page 6, L23-24: How the two weather stations (Qatraneh and Errabbah) represent climate conditions in the study area? If the stations are not representing actual conditions, the generated data (to fill the gaps in the recorded time series) will be subjective, and as a result poor hydrological performance will be obtained (it is seen also from the results).

   **AC:** It is common in hydrological modelling (and specifically using SWAT) to use data from nearby stations (Fuka et al., 2014) if they are reasonably close to the catchment because SWAT uses geographical weightage/interpolation when using data from nearby stations and each subbasin is linked to the station closest to its centroid. However, we agree with the Referee that this can be a source of uncertainty but in such data-challenging environment, these two stations were the closest with complete datasets that the model requires to run. This potential uncertainty emphasises the need to improve data monitoring (and accessibility) in these areas for better studies. The text is edited to point to this important note. Please see the modification to section 4.1 for a comment on the representativeness of stations.

3. **RC:** Page 6, Section 3.5: How the HRUs were defined? Please indicate if you used dominant or multiple (/threshold?).

   **Ac:** Section 3.5 is edited and threshold criteria defined.

4. **RC:** Page 7, Section 3.7: In this study the sediment yields are not measured but estimated using the streamflow data. This cause a subjective comparison results when testing different scenarios in this study: (i) It is obvious from the results that model scenarios that perform better in the simulation of streamflow, present higher performance in modeling sediment too. Therefore, any judgment on the performance of the input data in model simulation will be subjective. (ii) The inherent errors in the estimated sediment yields may be compromised/offset by the model prediction errors due to less quality input data, resulting in a wrong conclusion in scenario selection. The authors may provide a background on how the sediment data were estimated and how the above mentioned points may be justified when evaluating model-scenarios in this work.

**AC:** Please see the edits in section 3.7 for details of how sediment data were constructed. The equation provided was accredited and used to design the dam and manage the catchment. We hope it represents the conditions in the area (furthermore, it is the best available source for sediment data). However, we agree that there is dependency of the constructed sediment data on discharge data they correspond to, this may justify the correlation between model performance in simulating discharge and sediment.

The inputs used to simulate sediments are completely independent from the observation-constructed sediment data; therefore, we think that the goodness of fit of sediment simulation is properly assessed by comparing two independent series of data (sim & obs).

Quantitatively, the better statistics of sediment simulation could be a result of the nature and magnitude of sediment events compared to those of discharge and not because of the explained dependency. Given the complication of sediment simulation and data scarcity, we hope this reveals the Referee's concern and sheds light on the challenges facing data-poor areas.

5. **RC:** Page 7, L 29-31: The strong relationship between Q and T should be explained. The provided reference is not enough and not available for readers.

   **AC:** Appreciating the importance of this note, the equation used to construct sediment data is added (please see section 3.7). Howard Humphreys and Partners (1992) is a well-recognised consultant study undertaken for the purposes of designing the Wala dam and we are afraid it is beyond the scope of the current study to present the finer details of that study.

6. **RC:** There are many studies suggesting multi gauge evaluation (rather than single outlet comparison) in spatially explicit hydrological modeling, to prevent the spatial errors in the upstream catchments that may offset and not observed in the outlet. If the gauge data are not available in other tributaries of this study to conduct multi-gauge evaluation, at least the importance of this effort should be highlighted to also support the statement on Page 3, L3-5, on the investment efforts to improve the limited data.

   **AC:** We totally agree with this and recommend improving field measurements to provide trustworthy observed data. Unfortunately, we could hardly obtain observed data for the outlet (which is of high importance for being a dam location) and could not get hold of any further data (if any), otherwise multi-gauge evaluation would have been undertaken. This key recommendation is added to the conclusion (last two lines).

7. **RC:** Authors may add three other layers of data into Fig. 5: (i) river network; (ii) stream-flow station (Wadi Wala) where the models were evaluated; (iii) Wala dam. With the geographic coordinates only, it is hard to understand the system.

   **AC:** Please see Fig. 2 for better illustration of the system and magnified location of the dam. The Referee is also invited to watch the supplementary video of the Wala catchment for improved display and understanding of the system.

8. **RC:** Page 7, L27-28: The authors evaluated their model-scenarios using the discharge data of Wadi Wala station that is located downstream of the dam location. How the operation of dam and the effects on hydrological regime in downstream station was considered? Authors may explain if the operation of dam was simulated in the model.

   **AC:** True, but the dam was put in operation and impoundment started after 2002 and the measurements used were before that. Currently, there are discharge measurements upstream the

dam and managed water release through the dam tunnels to downstream areas (these will be used for future uses of the calibrated model). Please see the edited text for clarity.

9. **RC:** AC: Page 7, L32: The "parameterised" SWAT model is not clear. Does it mean that authors make change in model parameters (for any calibration purposes?). If so, it contradicts with several statements in the text that that the evaluations were performed prior to calibration.
   **AC:** Apologies, the term "parameterized" is removed to clear up confusion. We mean the model with its default parameters which are extracted from its original inputs.

10. **RC:** Page 7, L32: SWAT model operates on "daily" not monthly or yearly or seasonal. Please rephrase the sentence. Authors may aggregated data from daily to monthly etc.
    **AC** : True, SWAT considers daily basis but gives the option to produce monthly or yearly estimates. Please see the link below for definitions of SWAT inputs
    (http://swat.tamu.edu/media/69392/ch31_input_meas.pdf).

11. **RC:** Page 9, L7-8: "NSE drops ... and all cases". These are interesting results. Authors may explain why CFSR data performed better than local observations. Are the findings consistent with other studies around the world that applied CFSR in hydrological models?
    **AC:** Yes, similar results were found by several studies. Please see the added references recommending using the CFSR over local records (modified sections 4.1 and 4.2.2).

12. **RC:** Page 9, L11-13: usually a better performance is expected when using locally produced high resolution maps (e.g., the one from Al Bakri). Authors may explain the possible reasons why the course resolution map of global source performed better in this study.
    **Ac:**
    - Section 3.4.1 describes briefly the similarity between the three land use maps in showing two dominant types of vegetation and minor coverage by other land use classes.
    - Section 4.2.3 (1ˢᵗ paragraph) states that the performance of the best three scenarios (with the only difference being land use) is almost equal (just slight differences found).
    - Section 4.2.3 (2ⁿᵈ paragraph) explains the relatively little variation in spatial distribution and range of physical characteristics among the three land use maps and the close range of CN values (ranging from 80 to 84). In addition, it is suggested that "the method of HRU definition within SWAT selects the major land-use types in each HRU, thus potentially nullifying the gains of higher-resolution land-use maps with numerous smaller land-use classes", which means that the gain of having more minor details in Al-Bakri's map is not obvious in this specific case and the land use types suggested by the global map may provide slightly more accurate representation of the actual land use.

13. **RC:** Page 10, section Weather data: Please refer to my previous comment in this section (comment #2).
    **AC:** Please see response to comment #2 and the edited text in sections 4.2.2 and 4.2.4.

14. **RC:** Page 10, L12-16: (i) When looking at the top left graph in Fig. 9, the high performing statistics are usually representing low flows, while high flows are almost completely underestimated (or not predicted at all). How the authors will consider both low flows (important for drought) and high flows (major events and important for flooding and water saving) explain in their model evaluation

and scenario selection? (ii) As indicated in several places in the text, the Wala basin is located in semi-arid area that is prone to intense (sub-daily) rainfall events. My concern is that how model evaluation at the monthly scale will ensure representation of locally important short-term events? (iii) As mentioned previously, the data are compared at downstream station that might be affected by operation of a dam.

**AC:** i) We agree with the Referee's concern but as the statistics are unable to differentiate between low and high flows, the observed good relationship for scenario 16 with the observed at high flows is encouraging, whilst recognising that the simulated at times struggles to determine smaller events. Currently, we hope this fulfils the aim of this study and the next step will be an in-depth calibration to deal with low/high flows and preferentially alter their simulated values to match observed data. We noted an error in Figure 9 and have presented the correct figure.

(ii) We appreciate that ideally the use of daily information as both input and output would be preferable. Unfortunately, attempting to run the analysis at the daily temporal scale failed to achieve satisfactory results, this we feel could be the result of several factors (e.g. catchment lag, partial precipitation coverage of the catchment, data quality issues). The use of the monthly provided a more satisfactory output, we feel that this is the result of the temporally short (sub)daily precipitation being averaged over the month, removing the ephemeral nature of precipitation and subsequent river flows and sediment production. The original discharge measurements were daily and their monthly averages were calculated to match the simulation time interval. Please see the edits in paragraph 2 of section 3.7 and the text below as inserted at the end of the section:

"Whilst input data (climatic) are based at a daily temporal scale, the model outputs are considered at a monthly timescale for several reasons, i) daily observations of discharge and sediment are unavailable at the Wala station for the whole period of study, with only monthly observations available for model evaluation; ii) A shorter period (1990 – 1996) of daily observations are available at Wala station, but using these yields poor correlations <0.1 between daily model-simulated and observed discharge, iii) with incomplete/low quality measurements, potential for lag within the pairs of daily simulated and observed values (for model statistical evaluation) can present challenges, which can be reduced when using aggregated temporal data; and most importantly, iv) the objective of this study is to determine long-term flux within the catchment, avoiding the complexity presented by ephemeral systems and since the monthly comparison achieves reasonable fit between observed and simulated values, it is considered sufficient for evaluation of the current model with more convenience. However, all calculations of the model occur on a daily time step, which ensures that hydrological events are accounted for separately as they occur each day.

Similar approach is adopted in several studies, particularly using SWAT, in both humid and arid regions. Spruill et al. (2000) evaluate daily and monthly SWAT models simulation for a small watershed in central Kentucky and state that SWAT is an efficient tool for monthly runoff simulation with NSE values of 0.58 – 0.89 compared to - 0.04 – 0.19 for daily runoff simulation during the same period. The reason suggested is that the model poorly detects peak flows and recession rates while it performs better with total monthly values. For reasonable performance of SWAT, Huang and Zhang (2004) select to simulate discharge in a semi-arid catchment in China on a monthly basis, which leads to NSE of 0.88. The difference between daily and monthly simulations is investigated in watersheds of different scales by Heathman and Larose (2007) and the results show that simulating higher discharge rates, which is usually associated with larger watersheds, introduces greater uncertainty in SWAT discharge estimates and the study states that very good

model performance is achieved for monthly stream-flow estimation while the outputs of daily simulation are only within acceptable range."

(iii) Please see our response to comment (8): the dam was put into operation and impoundment started after 2002 and the measurements used were before that.

15. **RC:** Page 11, L13-22: Another reason is the use of statistics generated from inappropriate weather stations: please refer to my comment in this section (comment#2)
**AC:** This is a possibility and we are surely taking it into consideration. To check that, we tried to review the statistics generated for these stations and found them within expected ranges for the area. Please see response to comment 2 and the modified text in sections 4.2.2 and 4.2.4.

16. **RC:** Page 11, section 4.3: It would be interesting to apply the calibrated parameters of the model-scenarios to support the main idea of the study on how appropriate model setup and quality data help managers of data scarce watersheds when transferring knowledge (e.g., parameters) from neighboring watersheds. Please refer to my general comment.
**AC:** Please see our response to the General Comment. Apologies again for the confusion, the main idea is to reduce input uncertainty by using the best available datasets and invest on improving the sensitive ones (if required). However, the suggestion is interesting and it is already planned to use the calibrated model to support decision making in the area, for example the feasibility of raising the Wala dam which is being under investigation by the Jordanian Government currently. The calibrated model will be also used to suggest land management scenarios in the area but this is beyond the scope of this paper. Please also see the last two paragraphs of the conclusion which support the above.

**C. Technical corrections**

1. **RC:** Authors may keep the acronyms consistent: either provide full definition for all of the NSE, RSR, and PBIANS; with the acronyms in the bracket; or provide acronyms only.
**AC:** 'Nash-Sutcliffe Efficiency' removed in the abstract and added in the text. All consistent now (acronyms in the abstract and full definitions in section 3.7).

2. **RC:** Page 3, L3: "..quality and/or quality" : "..quality and/or quantity".
**AC:** Corrected.

3. **RC:** Page 2, L6: "..with extremely . . . and" is repetition of previous statement. Either remove this sentence or rephrase.
**AC:** Repeated statement removed. (Please note, the statement was in P3 not P2)

4. **RC:** Page 4, L16: "basin characteristics" are NOT represented by the USDA CN method, but the "surface runoff" is simulated.
**AC:** Sentence modified.

5. **RC:** Page 24, Fig. 5: for the ease of visual comparison, I suggest using similar coloring pattern for the same LU classes in the three maps.
   **AC:** New figure with similar coloring pattern inserted. (it is Fig. 3)

6. **RC:** Page 6, section 3.4.3: the authors may move this short paragraph to section 3.3, where you first discuss the DEM in the study area.
   **AC:** Paragraph moved as suggested.

7. **RC:** Page 7, L23: "Running SWAT . . .for full description": statement does not fit in this section. I suggest to move it to the section 3.2, where the authors introduce SWAT model, or to remove it.
   **AC:** Statement moved as suggested.

8. **RC:** Page 30, Fig. 9: Titles and legends are too small. Only scenario 16, and scenario 2 are presented in the figure, while the caption indicates scenarios 13, 18, 5, and 3 as well as 16, and 2. Please carefully check the figures, captions, and discussion in the text to match these three parts.
   **AC:** Apologies, a wrong figure was inserted, which is corrected now. It matches the text and captions.

9. **RC:** Page 30, Fig. 9: the explanation of the sediment yield graphs are missing from the caption and text.
   **AC:** Figure is corrected. Only discharge is meant to be displayed in Fig. 9.

10. **RC:** Page 31, Fig. 10: please provide high quality graphs with consistent size and formatting, and consistent scale in the vertical axis to help comparison of the graphs. With the current format it is hard to read and compare the graphs.
    **AC: W**e apologize for the inconvenience but this was felt to be the best way of presenting the images as a previous attempt to standardise the y-axis for all four scenarios made the figure difficult to read (and split into two pages), hence we kept the x-axis consistent and the background lines at a consistent interval in all figures (0.2) and we kindly ask the readers to consider the different scale of scenario 2 from the first 3 scenarios, which are presented on the same y-axis scale). However, if that still causes confusion, would you please consider the revised Figures 9 and 10 with all selected scenarios displayed with a consistent scale? We hope the comments of Referee #1 on the quality and illustrations support our response.

**References:**

[revised manuscript text omitted]

[1] (Van Griensven, 2005)

**Table 3. Results of parameters sensitivity analysis of scenario 16**

| Scenario No. | NSE (uncalibrated) | NSE (calibrated) |
|---|---|---|
| 16 | 0.56 | 0.64 |
| 2 | -12.00 | -11.29 |

**Table 4. Values of NSE calculated for uncalibrated and calibrated best and poorest-performing scenarios (16 and 2, respectively) for discharge simulation.**

**FIGURES**

[Figure]

**Figure 1. Location of the Wala Catchment.**

[Figure]

**Figure 2. The Wala catchment delineation into subbasins, stream pattern and the catchment outlet.**

[Figure]

**Figure 3. Land-use classification over the Wala catchment: a) Tarawneh (2007); b) Al-Bakri et al. (2013) ; c) WaterBase (2012). 1 SWAT land-use codes (Arnold et al., 2013).**

[Figure]

**Figure 4. Soil classification of the Wala catchment: a) WaterBase (2012); b) Tarawneh (2007).**

[Figure]

**Figure 5. Local weather stations, CFSR grid points; and local rainfall gauges with their percentage of missing data over the period 1971-2002.**

[Figure]

**Figure 6. (a) Daily minimum and maximum temperature of Qatraneh station; (b) Daily precipitation of Amman Airport gauge, for the period 2000 - 2003.**

[Figure]

**Figure 7. Values of $r^2$ calculated for prediction of a) discharge; b) sediment yield, from the 18 scenarios.**

[Figure]

**Figure 8. Values of PBIAS calculated for prediction of a) discharge; b) sediment yield, from the 18 scenarios.**

[Figure]

**Figure 9. Simulated and observed average monthly discharge (m³s⁻¹) for scenarios 2, 3, 5, 18, 13 and 16.**

[Figure]

**Figure 10. Simulated and observed monthly sediment yield (M-t/month) for scenarios 2, 5, 18 and 16.**

Scenario No.

[Figure]

Figure 9. Simulated and observed average monthly discharge ($m^3 s^{-1}$) for scenarios 16, 13, 18, 5, 3 and 2 (arranged as model performance descends downward).

[Figure]

Figure 10. Simulated and observed monthly sediment yield (M t/month) for scenarios 16, 18, 5 and 2 (arranged as model performance descends downward).

[Figure]

**Figure 11. Workflow illustrating a generic pre-calibration approach based on the methodology outlined in this study.**

---

## Referee Report (RR1)

**Review of the "A pre-calibration approach to select optimum inputs for hydrological models in data-scarce regions"**

Regarding the use of monthly data for model evaluation, rather than daily:

It is standard practice that large scale watersheds are evaluated based on monthly rather than daily data. This is mainly due to: (1) larger errors associated with the data at daily time step that are usually compromised when aggregating to monthly; (2) assumptions and simplifications in large scale models that are not designed to represent very detail small temporal and spatial scale processes.

As I mentioned previously (comment#14, below please find it), the Wala basin is located in semi-arid part of the world; and the short-term events (intense daily or sub-daily rainfall and floods) are major concern. While the model is evaluated at monthly, the daily (and sub-daily) events may be under estimated. The authors have not addressed or discussed this issue in their article. I strongly recommend to report this in the paper at least in the discussion part. I understand the main goal of the paper is to address the importance of data monitoring, etc., but even in that case the author can argue this issue to support their goal and objectives.

> *"my previous comment #14: Page 10, L12-16: (i) When looking at the top left graph in Fig. 9, the high performing statistics are usually representing low flows, while high flows are almost completely underestimated (or not predicted at all). How the authors will consider both low flows (important for drought) and high flows (major events and important for flooding and water saving) explain in their model evaluation and scenario selection? (ii) As indicated in several places in the text, the Wala basin is located in semi-arid area that is prone to intense (sub-daily) rainfall events. My concern is that how model evaluation at the monthly scale will ensure representation of locally important short-term events? (iii) As mentioned previously, the data are compared at downstream station that might be affected by operation of a dam."*

Page 5, Line 12-13: Repeated statement. Please remove one.

Page 9, Line 29-30: As already mentioned in my previous review, you cannot run the SWAT model on monthly basis. It operates on daily time step, and it just prints the outputs on monthly or yearly formats depending on the user's need. Please rewrite the statement (all SWAT users are quite familiar with the model operation, may be no need to mention how you ran the model but refer to post processing of the data only).

---

## Author Response (AR2)

**Ms Esra'a Tarawneh**

School of Engineering
The Quadrangle
The University of Liverpool
Brownlow Hill
Liverpool L69 3GH

E erat@liverpool.ac.uk

www.liv.ac.uk

3rd October 2016

Dear Editor,

**Re. Resubmission of manuscript HESS-2016-242**

We are pleased to resubmit for publication the revised version of manuscript HESS-2016-242 "*A pre-calibration approach to select optimum inputs for hydrological models in data-scarce regions.*" We have received with pleasure and highly appreciate the positive feedback and improvements suggested by the Editor and Referee #2.

Please find enclosed with this letter a revised manuscript and later in this document, the associated Authors' response and a marked-up manuscript in respect of our resubmission of the above paper.

We have taken this precious opportunity to revise the manuscript in accordance with the latest suggestions received and hopefully, our response addresses the concern raised through the review process resulting in a paper which will be acceptable for publication.

Our justifications for publication in *Hydrology and Earth System Sciences,* assertions of significance and confirmations in respect of authorship and prior publication remain as per our original submission dated 23rd May 2016.

Yours sincerely,

Ms Esra'a Tarawneh

on behalf of co-authors Dr Jonathan Bridge and Dr Neil Macdonald.

**Authors' Response to the Review Comments**

**Editor (Dr Elena Toth) – Received 30[th] Sep 2016**

**EC:**

Dear Authors,

Thank you indeed for the revised version where you addressed the remaining comments.

Referee#2 has reviewed the manuscript but had an intermediate version of the revision and you may therefore discard the last two comments (on page 5 and page 9: they refer to an older version and have been already amended in the paper).

On the other hand I agree with the Referee that you should consider to add in the conclusions a comment on the fact that while the model is evaluated at monthly, the daily (and sub-daily) events may be under estimated, and that this issue is very important in semi-arid basins.

I look forward to reading the final version.

My warmest wishes,

Elena Toth
HESS Editor

**AC:** Once again, would like to thank the Editor, Dr Toth, for handling the review process and sharing positive feedback with useful suggestions. We appreciate the invitation to resubmit the final manuscript and hope our revision addresses any remaining concerns of the Editor.
Please see the addition at the end of the Conclusions section in response to the point raised above regarding the use of monthly/daily events for model evaluation.

**Anonymous Referee #2 - Submitted 28[th] Sep 2016**

**1. RC:** Regarding the use of monthly data for model evaluation, rather than daily: It is standard practice that large scale watersheds are evaluated based on monthly rather than daily data. This is mainly due to: (1) larger errors associated with the data at daily time step that are usually compromised when aggregating to monthly; (2) assumptions and simplifications in large scale models that are not designed to represent very detail small temporal and spatial scale processes. As I mentioned previously (comment#14, below please find it), the Wala basin is located in semi-arid part of the world; and the short-term events (intense daily or sub-daily rainfall and floods) are major concern. While the model is evaluated at monthly, the daily (and sub-daily) events may be under estimated. The authors have not addressed or discussed this issue in their article. I strongly recommend to report this in the paper at least in the discussion part. I understand the main goal of the paper is to address the importance of data monitoring, etc., but even in that case the author can argue this issue to support their goal and objectives.

"my previous comment #14: Page 10, L12-16: (i) When looking at the top left graph in Fig. 9, the high performing statistics are usually representing low flows, while high flows are almost completely underestimated (or not predicted at all). How the authors will consider both low flows (important for drought) and high flows (major events and important for flooding and water saving) explain in their model evaluation and scenario selection? (ii) As indicated in several places in the text, the Wala basin is located in semi-arid area that is prone to intense (sub-daily) rainfall events. My concern is that how model evaluation at the monthly scale will ensure representation of locally important short-term events? (iii) As mentioned previously, the data are compared at downstream station that might be affected by operation of a dam."

**1. AC:** The Authors would like to express thankfulness to Referee #2 for reviewing the revised manuscript and suggesting further improvements. It is a pleasure that our responses to the previous review could clear up the majority of the Referee's concerns and we hope the current version will address the remaining and meet the Editor's acceptance for publication in the HESS.

In recognition of the importance of the point raised by the Referee regarding the use of monthly data for model evaluation in semi-arid regions, we have added a comment in the conclusion to notify readers that by using monthly data to evaluate the case study model, we do not discard the high importance of temporal resolution of data in semi-arid regions and the possibility that daily/sub-daily events could be underestimated if coarser resolution (e.g. monthly) is used. We also stress on the need to improve data monitoring in these areas to involve more detailed measurements that can be used to set up and evaluate models. Please see the modified conclusion in the revised manuscript in addition to our previous response to comment #14 of Referee #2.

**2. RC:** - Page 5, Line 12-13: Repeated statement. Please remove one.
  - Page 9, Line 29-30: As already mentioned in my previous review, you cannot run the SWAT model on monthly basis. It operates on daily time step, and it just prints the outputs on monthly or yearly formats depending on the user's need. Please rewrite the statement (all SWAT users are quite familiar with the model operation, may be no need to mention how you ran the model but refer to post processing of the data only).

**2. AC:** The two comments above have been addressed through the previous review and accepted by the Editor.

[revised manuscript text omitted]

Tachikawa, T.*, et al.*, Characteristics of aster gdem version 2. ed. *Geoscience and Remote Sensing Symposium (IGARSS), 2011 IEEE International*, 2011, 3657-3660.

Tarawneh, E., 2007. *Water and sediment yield for wala dam catchment area.* Thesis (MSc). Mutah University.

Tessema, S. 2011. Hydrological modeling as a tool for sustainable water resources management: A case study of the awash river basin.

Tingsanchali, T. and Gautam, M. R. 2000. Application of tank, nam, arma and neural network models to flood forecasting. *Hydrological Processes,* 14(14), 2473-2487.

Trondalen, J. M., 2009. Climate changes, water security and possible remedies for the middle east. *United Nations Educational, Scientific and Cultural Organization.*

Usda, S. 1972. National engineering handbook, section 4: Hydrology. *Washington, DC*.

Usda, S. 1986. Urban hydrology for small watersheds. *Technical release,* 55, 2-6.

Usepa, 2002. *Guidance for quality assurance project plans for modeling.* Washington, D.C.: USEPA, Office of Environmental Information, Report EPA/240/R-02/007.

Van Griensven, A. 2005. Sensitivity, auto-calibration, uncertainty and model evaluation in swat2005. *Unpublished report*.

Wala Dam Management, 2013. Field visit to the wala catchemment,. *In:* Bridge, J. and Tarawneh, E. eds.

Wang, W.*, et al.* 2011. An assessment of the surface climate in the ncep climate forecast system reanalysis. *Climate dynamics,* 37(7-8), 1601-1620.

Waterbase, 2012. *Global landuse and soil maps* [online]. Available from: http://www.waterbase.org/home.html [Accessed 20 November 2013].

Wheater, H., Sorooshian, S. and Sharma, K., 2008a. *Hydrological modelling in arid and semi-arid areas.* Cambridge University Press.

Wheater, H., Sorooshian, S. and Sharma, K., 2008b. Modelling hydrological processes in arid and semi-arid areas: An introduction to the workshop. *Hydrological Modelling in Arid and Semi-Arid Areas.(Cambridge University Press) Cambridge.*

Williams, J. 1969. Flood routing with variable travel time or variable storage coefficients. *Trans. ASAE,* 12(1), 100-103.

Williams, J. 1980. Spnm, a model for predicting sediment, hosphorous and nitrogen yields from agricultural basins *Journal of the American Water Resources Association,* 16(5), 843-848.

Williams, J. R. and Singh, V., 1995. *The epic model: Computer models of watershed hydrology.* USA: Water Resources Publications.

Wischmeier, W. and Smith, D., 1965. Usda agriculture handbook 282. Washington DC, US Department of Agriculture.

Wolffc, I. K. 2011. Land-use change scenarios for the jordan river region. *Int. J. of Sustainable Water and Environmental Systems,* 3(1), 25-31.

Wood, M. K. and Blackburn, W. H., 1984. An evaluation of the hydrologic soil groups as used in the scs runoff method on rangelands1. Wiley Online Library.

Wu, Y.*, et al.* 2016. Evaluation of the applicability of the swat model in an arid piedmont plain oasis. *Water Science and Technology,* 73(6), 1341-1348.

Yen, B. 1995. Criteria for evaluation of watershed models. *Journal of Irrigation and Drainage Engineering,* 121(1), 130-131.

Zhang, L.*, et al.* 2016. Comparison of swat and dlbrm for hydrological modeling of a mountainous watershed in arid northwest china. *Journal of Hydrologic Engineering,* 21(5), 04016007.

Zhang, X., *et al.* 2005. Trends in middle east climate extreme indices from 1950 to 2003. *Journal of Geophysical Research: Atmospheres,* 110(D22), D22104.

| Scenario No. | Land-use map[1] | Soil map[2] | Weather data | Madaba station | No. of HRUs | NSE | RSR |
|---|---|---|---|---|---|---|---|
| 16 | c | b | CFSR | – | 47 | 0.56 | 0.66 |
| 10 | b | b | CFSR | – | 67 | 0.56 | 0.67 |
| 4 | a | b | CFSR | – | 68 | 0.55 | 0.67 |
| 13 | c | a | CFSR | – | 48 | -0.32 | 1.15 |
| 7 | b | a | CFSR | – | 63 | -0.36 | 1.17 |
| 1 | a | a | CFSR | – | 60 | -0.36 | 1.17 |
| 18 | c | b | Local | Excluded | 47 | -0.36 | 1.17 |
| 12 | b | b | Local | Excluded | 67 | -0.43 | 1.19 |
| 6 | a | b | Local | Excluded | 68 | -0.69 | 1.30 |
| 17 | c | b | Local | Included | 47 | -2.90 | 1.97 |
| 11 | b | b | Local | Included | 67 | -3.16 | 2.04 |
| 5 | a | b | Local | Included | 68 | -3.56 | 2.13 |
| 15 | c | a | Local | Excluded | 48 | -4.69 | 2.38 |
| 9 | b | a | Local | Excluded | 63 | -4.84 | 2.42 |
| 3 | a | a | Local | Excluded | 60 | -5.39 | 2.53 |
| 14 | c | a | Local | Included | 48 | -11.25 | 3.50 |
| 8 | b | a | Local | Included | 63 | -11.42 | 3.52 |
| 2 | a | a | Local | Included | 60 | -12.00 | 3.61 |

Table 1. Number of HRUs and values of NSE and RSR calculated for 18 scenarios for comparison of observed and simulated average monthly discharge ($m^3s^{-1}$) at the Wala catchment outlet. 1 Land-use maps: a) Tarawneh (2007); b) Al-Bakri et al. (2013); c) WaterBase (2012). 2 Soil maps: a) WaterBase (2012); b) Tarawneh (2007).

| Scenario No. | Land-use map[1] | Soil map[2] | Weather data | Madaba station | No. of HRUs | NSE | RSR |
|---|---|---|---|---|---|---|---|
| 16 | c | b | CFSR | _ | 47 | 0.79 | 0.46 |
| 10 | b | | CFSR | _ | 67 | 0.66 | 0.58 |
| 4 | a | | CFSR | _ | 68 | 0.60 | 0.64 |
| 18 | c | | Local | Excluded | 47 | -0.11 | 1.06 |
| 12 | b | | | | 67 | -0.11 | 1.06 |
| 17 | c | | Local | Included | 47 | -1.67 | 1.63 |
| 11 | b | | | | 67 | -1.81 | 1.68 |
| 6 | a | | Local | Excluded | 68 | -2.97 | 1.99 |
| 5 | a | | Local | Included | 68 | -7.21 | 2.86 |
| 13 | c | a | CFSR | _ | 48 | -12.74 | 3.71 |
| 7 | b | | CFSR | _ | 63 | -16.47 | 4.18 |
| 1 | a | | CFSR | _ | 60 | -22.70 | 4.87 |
| 15 | c | | Local | Excluded | 48 | -26.72 | 5.26 |
| 9 | b | a | | | 63 | -36.01 | 6.08 |
| 14 | c | | Local | Included | 48 | -42.16 | 6.57 |
| 8 | b | | | | 63 | -48.98 | 7.07 |
| 3 | a | | Local | Excluded | 60 | -59.72 | 7.79 |
| 2 | a | | Local | Included | 60 | -85.06 | 9.28 |

Table 2. Number of HRUs and values of, NSE and RSR calculated for 18 scenarios for comparison of observed and simulated average monthly sediment yield (t/month) at the Wala catchment outlet. 1 Land-use maps: a) Tarawneh (2007); b) Al-Bakri et al. (2013); c) WaterBase (2012). 2 Soil maps: a) WaterBase (2012); b) Tarawneh (2007).

| Name | Description[1] | Rank |
|------|----------------|------|
| CN2 | Initial SCS CN II value (Curve Number) | 1 |
| SOL_AWC | Available water capacity (mm $H_2O$/mm soil) | 2 |
| SOL_Z | Soil depth (mm) | 3 |
| SURLAG | Surface runoff lag time (days) | 4 |
| ESCO | Soil evaporation compensation factor | 5 |
| CH_N | Manning's n value for main channel | 6 |
| ALPHA_BF | Baseflow alpha factor [days] | 7 |

[1] (Van Griensven 2005)

**Table 3. Results of parameters sensitivity analysis of scenario 16**

| Scenario No. | NSE (uncalibrated) | NSE (calibrated) |
|---|---|---|
| 16 | 0.56 | 0.64 |
| 2 | -12.00 | -11.29 |

**Table 4. Values of NSE calculated for uncalibrated and calibrated best and poorest-performing scenarios (16 and 2, respectively) for discharge simulation.**

**FIGURES**

[Figure]

**Figure 1. Location of the Wala Catchment.**

[Figure]

**Figure 2. The Wala catchment delineation into subbasins, stream pattern and the catchment outlet.**

[Figure]

**Figure 3. Land-use classification over the Wala catchment: a) Tarawneh (2007); b) Al-Bakri et al. (2013) ; c) WaterBase (2012). 1 SWAT land-use codes (Arnold et al. 2013).**

[Figure]

**Figure 4. Soil classification of the Wala catchment: a) WaterBase (2012); b) Tarawneh (2007).**

[Figure]

**Figure 5. Local weather stations, CFSR grid points; and local rainfall gauges with their percentage of missing data over the period 1971-2002.**

[Figure]

**Figure 6.** **(a) Daily minimum and maximum temperature of Qatraneh station; (b) Daily precipitation of Amman Airport gauge, for the period 2000 - 2003.**

[Figure]

**Figure 7. Values of $r^2$ calculated for prediction of a) discharge; b) sediment yield, from the 18 scenarios.**

[Figure]

**Figure 8. Values of PBIAS calculated for prediction of a) discharge; b) sediment yield, from the 18 scenarios.**

[Figure]

**Figure 9. Simulated and observed average monthly discharge (m³s⁻¹) for scenarios 2, 3, 5, 18, 13 and 16.**

[Figure]

**Figure 10. Simulated and observed monthly sediment yield (M-t/month) for scenarios 2, 5, 18 and 16.**

[Figure]

**Figure 11. Workflow illustrating a generic pre-calibration approach based on the methodology outlined in this study.**